# Geomorphology, Geoarchaeology, and Geochronology of the Upper Pleistocene Archaeological Site of El Olivo Cave (Llanera, Asturias, Northern Spain)

**Jesús F. Jordá Pardo** [1,2,*], **David Álvarez-Alonso** [2,3], **María de Andrés-Herrero** [2,3], **Daniel Ballesteros** [4], **Pilar Carral** [5], **Aitor Hevia-Carrillo** [6], **Jorge Sanjurjo** [7], **Santiago Giralt** [8] and **Montserrat Jiménez-Sánchez** [9]

[1] Department of Prehistory and Archaeology, Faculty of Geography and History, UNED, Senda del Rey 7, E-28040 Madrid, Spain

[2] Prehistoric Archaeology Research Group—GIAP, Complutense University of Madrid, C. Profesor Aranguren s/n, Ciudad Universitaria, E-28040 Madrid, Spain; david.alvarez@ucm.es (D.Á.-A.); maria.deandres@ucm.es (M.d.A.-H.)

[3] Departament of Prehistory, Ancient History and Archaeology, Complutense University of Madrid, C. Profesor Aranguren s/n, Ciudad Universitaria, E-28040 Madrid, Spain

[4] Department of Earth Sciences and Condensed Matter Physics, University of Cantabria, Avenida de los Castros s/n, E-39005 Santander, Spain; ballesterosd@unican.es

[5] Department of Geology and Geochemistry, UAM, Campus de Cantoblanco, E-28049 Madrid, Spain; pilar.carral@uam.es

[6] EID-UNED, Department of Prehistory and Archaeology, Faculty of Geography and History, UNED, Senda del Rey 7, E-28040 Madrid, Spain; aitorhevia@gmail.com

[7] University Institute of Geology Isidro Parga Pondal, University of A Coruña Campus de Elviña s/n, E-15011 A Coruña, Spain; jorge.sanjurjo.sanchez@udc.es

[8] Geoscience Barcelona Institut (CSIC), C. Lluís Solé i Sabarís s/n, E-08028 Barcelona, Spain; sgiralt@ictja.csic.es

[9] Grupo Geomorfología y Cuaternario, Departamento de Geología, Universidad de Oviedo, C. Jesús Arias de Velasco s/n, E-33005 Oviedo, Spain; mjimenez@uniovi.es

[*] Correspondence: jjorda@geo.uned.es

**Abstract:** El Olivo Cave (Pruvia de Arriba, Llanera, Asturias, Spain) is a small karst cave located in the Aboño River basin and formed in the Cretaceous limestone of the Mesozoic cover of the Cantabrian Mountains (north of the Iberian Peninsula). It contains an important upper Pleistocene sedimentary, archaeological, and paleontological record, with abundant technological evidence and faunal remains. The archaeological record shows a first occupation that could correspond to the Middle Paleolithic and a second occupation in the Middle Magdalenian. The stratigraphic sequence inside and outside the cave was studied with geoarchaeological methodology. In this paper, the lithostratigraphic sequence is analyzed, and the data from the granulometric, mineralogical, edaphological, and radiometric analyses are presented. The results of these analyses enable an accurate interpretation of both the lithostratigraphy of the deposit and the processes responsible for its formation and subsequent evolution. The available numerical dates allow us to locate the first sedimentation episode in the cave in OIS 7a, in the Middle Pleistocene, the base of the outer fluvial sedimentation in the cold OIS 3a stage of the Upper Pleistocene and the Magdalenian occupation in the Last Glacial Maximum (OIS 2) at the end of the Late Pleistocene.

**Keywords:** karst; cave sediments; upper Pleistocene; Cantabrian zone; Northern Iberia

## 1. Introduction

El Olivo Cave is situated in the Cantabrian Region, northern Iberian Peninsula (Figure 1A), at UTM coordinates (Zone 30 N; ETRS89) X. 275,133 Y. 4,815,338, and an altitude of 145 m a.s.l. The cave is 10 m long and has an entrance 2 × 3 m in diameter. It formed in Mesozoic limestone. The cave is close to the Cabornio stream, belonging to the Aboño River basin (123 km$^2$ in size), which flows towards the N (Figure 1).

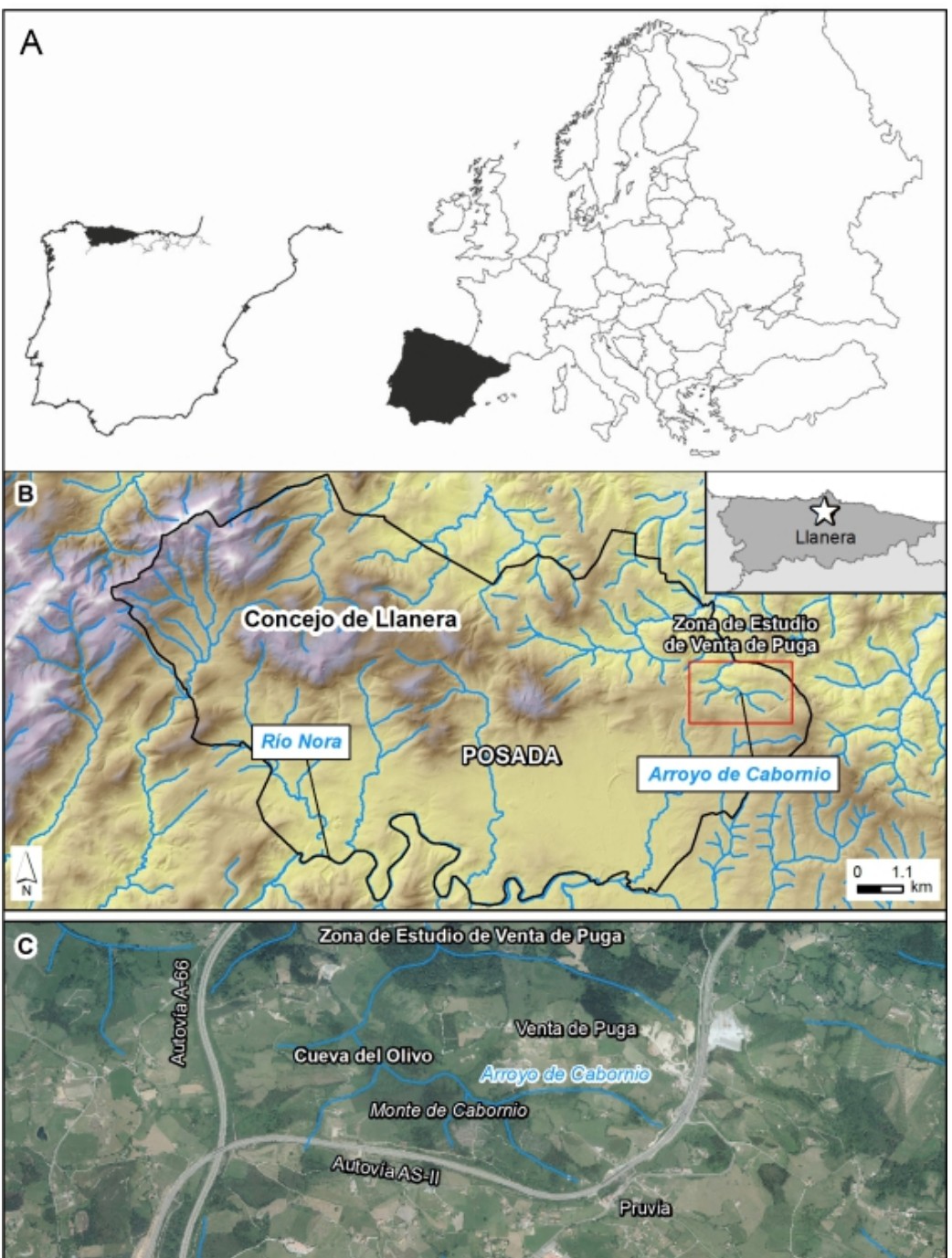

**Figure 1.** (**A**) Location of El Olivo Cave in the northern Iberian Peninsula. (**B**) Geological areas in the surroundings of the cave, which is in the Aboño River basin. Geology after. The red rectangle indicates the studied area of Venta de Puga. (**C**) Orthophotography of the Venta de Puga study area, where the El Olivo Cave is located.

The cave, well known by the locals, was used as a refuge during the Spanish Civil War (1936–1939). Even though a speleological group explored and inventoried El Olivo Cave in 1985, an archaeological exploration of the cave was never carried out. In July 2012, a survey was carried out in El Olivo due to the excavation of the open-air Mousterian site of El Barandiallu, located about 3 km to the west of the cave [1]. In this first year, a test pit was excavated to verify the existence of a Paleolithic deposit. Between 2013 and 2017, systematic archaeological fieldwork was conducted in the cave (Figures 2 and 3) [1–5].

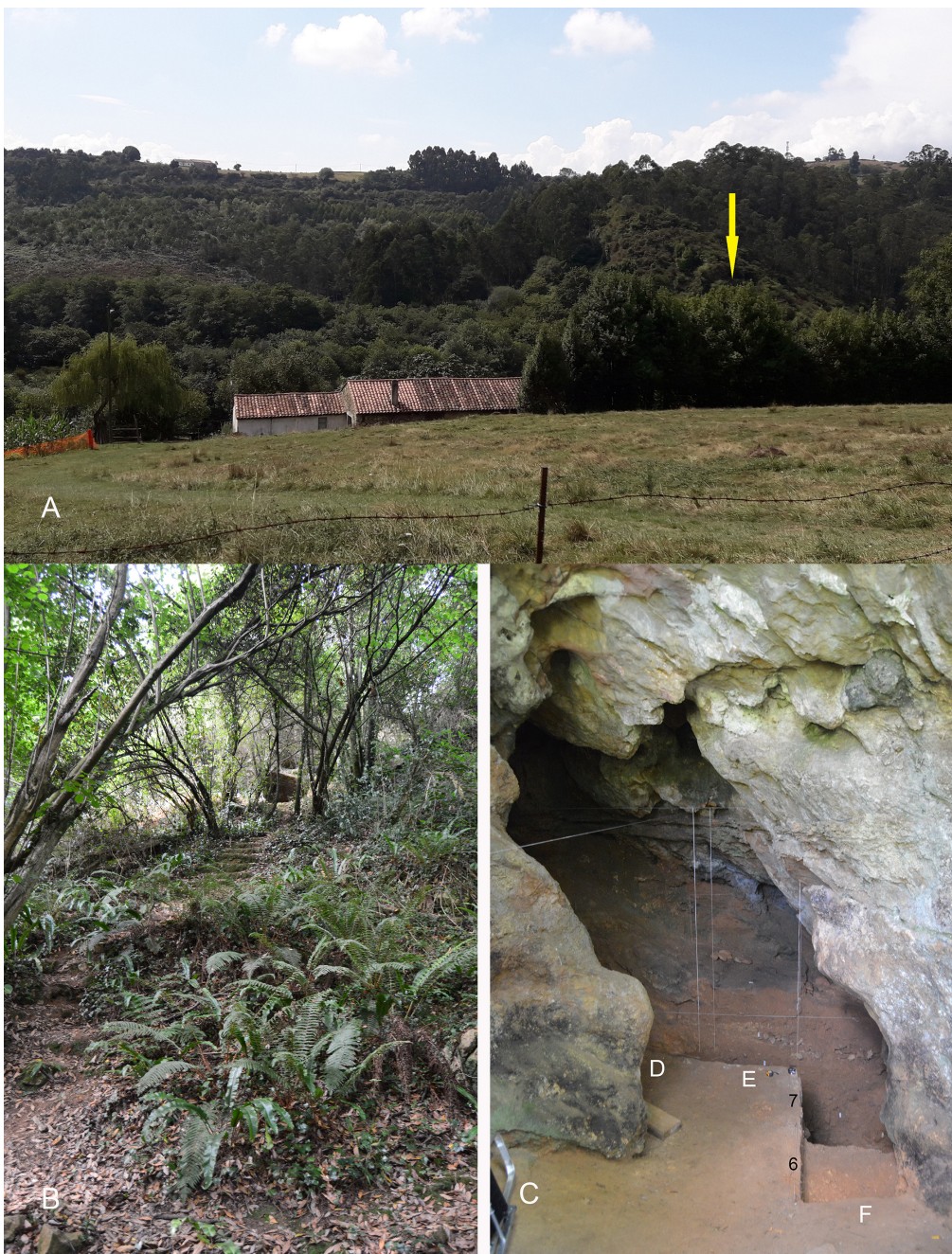

**Figure 2.** General view with the location of El Olivo Cave ((**A**), yellow arrow), detail of its access (**B**), and interior of the cave showing the excavated areas (the numbers 6 and 7 and capital letters D, E and F indicate the names of the excavation squares) (**C**).

In this paper, we present the results of the geoarchaeological investigations carried out in El Olivo Cave, which reveal complex fluviokarstic evolution of its sedimentary fill, at the same time that they show important action of external geomorphological agents during the Last Glacial Maximum (LGM) and the Late Glacial period in this sector of the northern coastal strip of the Iberian Peninsula, as also occurs in other deposits with similar chronology, such as Bañugues (Gozón, Asturias) [6,7] and El Barandiallu (Llanera, Asturias) [1,8,9].

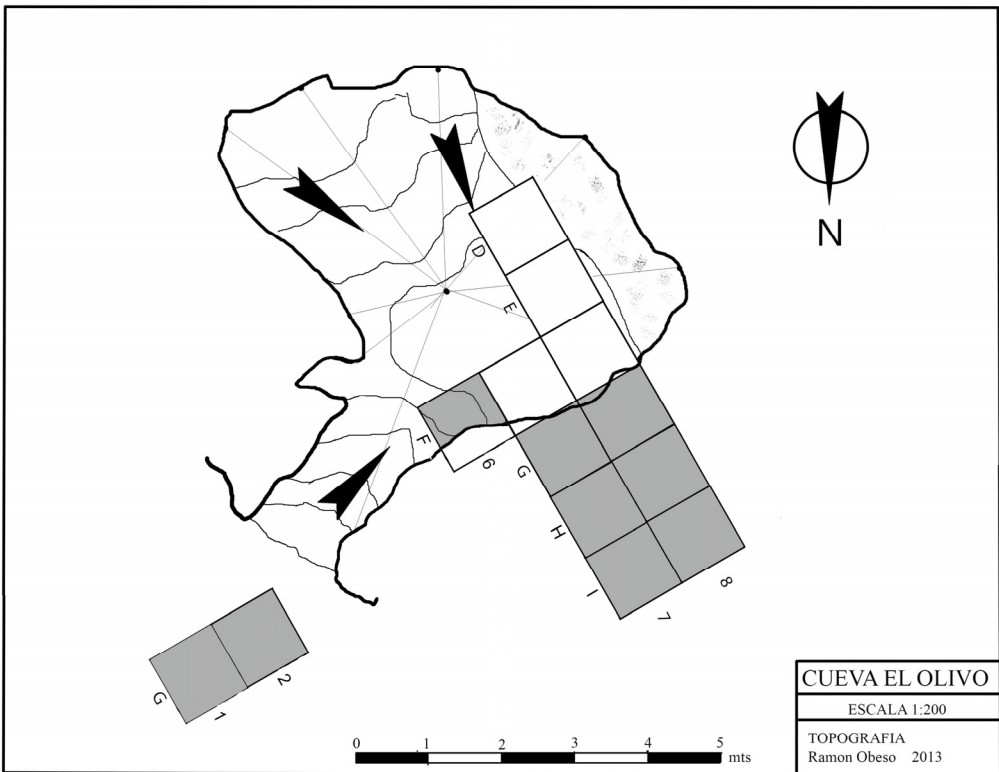

**Figure 3.** El Olivo Cave plan [4], with indication of the excavation grid and the excavated squares (in gray) (numbers and capital letters indicate the names of the excavated squares; black arrows indicate slopes).

The objective of this work is threefold: (i) to develop the evolutionary geoarchaeological model of El Olivo Cave karst and its surroundings, (ii) to characterize the archaeological site geoarchaeologically, and (iii) to establish the chronostratigraphy of its deposits. This geoarchaeological study addresses the following aspects:

Study of the sedimentary sequence of the archaeological site;

Interpretation of the formation and transformation processes that gave rise to the current configuration of its archaeological record;

Differentiation, to the extent possible, of natural processes (N transforms) and/or cultural processes of anthropic origin (C transforms) [10];

Identification of sedimentary processes;

Identification of diagenetic and postdepositional processes [11];

Establishment of the geoarchaeological evolution of the site.

## 2. Materials and Methods

The applied work methodology combined geological, geomorphological, geoarchaeological, and geochronological techniques to establish the geomorphological evolution of El Olivo Cave and its surroundings and the archaeosedimentary fill at the site.

### 2.1. Geomorphology

To understand the origin and development of El Olivo Cave and the local landscape evolution during the Pleistocene, the geomorphology of the cave and its surroundings were mapped using a combination of photo interpretation, GIS tools, and fieldwork in ArcGIS 10.2 [12]. Specifically, the cave plan was created following the method outlined by Ballesteros et al. [13], which involves collecting survey data using the DistoX2 laser range-finder [14] and processing it in Compass 5.09 software [15].

### 2.2. Lithostratigraphic Study

The geoarchaeological study of the site was carried out through the detailed analysis and lithostratigraphic description of the archaeosedimentary sequences obtained in the archaeological excavations located inside and outside the cave. These sequences were also appropriately sampled for sedimentological and edaphological analyses, the results of which are detailed below.

### 2.3. Sedimentological Analyses

Sedimentological analyses carried out involved the textural classification of the sediments using:

Laser granulometry for the fraction finer than 2 mm;
Phi granulometry for the total sediment, including the coarse section;
Mineralogical identification using X-ray diffraction (XRD) of the fraction finer than 0.63 mm.

These analyses were conducted at the Assistance Center for Research in Geological Techniques at the Complutense University of Madrid.

The granulometric analysis was carried out following this protocol:

Suspension of a known quantity of each of the samples;
Sample disintegration;
Sieving at 700 μm, the upper limit of technical measurement capacity in laser granulometric equipment;
Phi scale granulometry of the fractions greater than 700 μm with the 4, 2, and 1 mm mesh size sieves;
Laser granulometry for fractions finer than 700 μm.

The laser granulometric analysis was carried out with Honeywell Microtrac x100 equipment, with the capacity to measure fractions from 700 to 0.10 μm. The statistical treatment of the granulometry data was performed with the GRADISTAT software version 8.0 [16,17], which allows the sediment in the detrital samples to be grouped statistically into different textural groups depending on the greater or lesser presence of gravel, sand, and mud (silt and clay). To do this, we have applied the granulometric classification adopted by the program that comes from the modification of the Udden (1914) [18] and Wentworth (1922) [19] scales: pebbles (clasts with diameters greater than 64 mm), gravel (clasts with diameters between 64 mm and 2 mm), sand (grains between 2 mm and 62 microns), silt (grains between 62 and 2 microns), and clay (particles smaller than 2 microns).

A Brucker D8 ADVANCE model diffractometer was used to obtain X-ray diffraction data. Disoriented dust diffraction diagrams to characterize the mineralogy of the total sample were obtained in an angular interval from 2 to 65°, a step size of 0.02°, and a step time of 1 s. Using Brucker's EVA software, the semiquantitative analysis followed Chung's (1975) [20] method.

### 2.4. Soil Analyses

Soil analyses carried out in the Laboratory of Edaphology in the Department of Geology and Geochemistry at the Autonomous University of Madrid consisted of determining color, pH, total carbonates -$CaCo_3$-, organic matter -OM-, organic charcoal -OC-, electrical conductivity, salts, cation concentration, and osmotic pressure.

For the precise determination of the color -dry- the Munsell Soil Color Charts [21] were used as a reference for an objective description and a standardized denomination. For the color description, these tables use three basic parameters expressed in the following order: hue + value + chroma.

The pH was measured following the Soil Science Society of America criteria [22] from a soil:water = 1:2.5 ratio to obtain the current acidity since the soil, given its dynamics, is very sensitive to changes in its evolution. One of the related factors to such modifications is the hydrogen ion concentration. The concept of pH used here is the same as for true solutions

despite being a heterogeneous system. Its measurement is a technique that, despite its simplicity, acquires very useful routine data.

The addition of a known amount of acid that causes the dissolution of the carbonates and the subsequent titration of the excess of added acid (acid not consumed) with a base was used for the quantitative determination of inorganic charcoal. The primary reaction is $CaCO_3 + 2HCl \rightarrow CaCl_2 + CO_2 + H_2O$.

The determination of total charcoal, including different forms of C presentation such as carbonates, condensed forms, plant residues, etc., was calculated using the easily oxidizable organic matter. For this reason, its determination was made via wet oxidation of organic C by the excess of potassium dichromate in a strongly sulfuric medium, using the dilution heat of this acid to facilitate oxidation according to the formula $3C + 2K_2Cr_2O_7 + 8H_2SO_4 \rightarrow 2K_2SO_4 + 2Cr_2(SO_4)_3 + 3CO_2 + 8H_2O$. Excess of dichromate is titrated with ferrous ammonium sulfate, $(NH_4)_2Fe(SO_4)_2 \cdot 6H_2O$, Mohr's salt, in the presence of phosphoric acid, using diphenylamine as the indicator.

*2.5. Geochronology*

Three different ages were determined using a speleothem, a cave fluvial deposit, and a paleontological remain to establish the chronological framework for El Olivo Cave. Flowstone on the cave walls was selected for dating using alpha-spectrometry at the U/Th Geoscience Barcelona Institute-CSIC (Spain) laboratory. The flowstone was identified as a perched ledge located 2.05 m above the cave floor (before archaeological excavation). To obtain a sample for dating, the OL-3 sample was collected using a hammer and chisel, and approximately 20 g of carbonate powder was extracted using a hand diamond drill in the laboratory. The U and Th isotopes separation and purification procedures followed the Bischoff et al. (1988) method [23] and the isotope electrodeposition method developed by Talvitie (1972) [24], modified by Hallstadius (1984) [25]. The radioisotope concentrations were determined using an ORTEC OCTETE PLUS spectrometer equipped with eight BR-024-450-100 detectors developed by Ivanovich and Harmon (1992) [26]. The speleothem age was calculated based on the time of analysis (the year 2017) using the Rosenbauer (1991) method [27] and expressed in years before the time of analysis with an uncertainty error of two standard deviations (2σ).

The sample OL-4 of cave fluvial deposits was dated using optically stimulated luminescence (OSL) at the Institute of Geology Isidro Parga Pondal, University of Coruña (Spain). To extract the sample, an opaque PVC tube (20 cm long, 55 mm diameter, and 4 mm wall) was driven into a homogeneous quartz-rich sand layer (OL.Exterior.3) at a depth of 0.97 m. The tube was immediately covered with duct tape foil to prevent light exposure and preserve humidity and sediment deformation. In the laboratory, samples were processed under red light conditions. The sand in the tubes was dried, sieved, and treated with HCl and $H_2O_2$ to remove potential carbonate and organic matter. Quartz was separated from feldspar and heavy minerals using centrifugation and diluted in HF to obtain pure quartz. The purity of the quartz was verified using infrared stimulated luminescence (IRSL), and preheat and bleached-aliquot recovery tests were carried out following the methods of Murray and Wintle (2003) [28]. OSL signals were recorded using an automated RISØ TL/OSL-DA-15 reader equipped with a photomultiplier EMI 9635 QA (PMT) and a $^{90}Sr/^{90}Y$ source. A 6 mm thick Hoya U-340 filter was placed between the aliquots and the PMT to measure the UV range emission. The single-aliquot regeneration (SAR) protocol of Murray and Wintle (2000) [29] was applied to 35 multigrain aliquots to estimate the equivalent dose (De) using the central age model [30] to date the sample. The activity of radioisotopes ($^{40}K$, $^{238}U$, $^{235}U$, and $^{232}Th$) was inferred using low background gamma-ray spectrometry. Calcined and ground sediments were measured using a coaxial Canberra XTRA gamma detector (Ge Intrinsic) model GR6022 within a 10 cm thick lead shield. The alpha dose rate was neglected due to the HF etching step, the beta dose rate was corrected [31], and the cosmic dose rate was estimated using the methods of Prescott

and Hutton (1994) [32] and Guerin et al. (2011) [33]. The OSL age is expressed in years before the time of analysis (the year 2009) with a corresponding 2σ uncertainty.

The third date was obtained through the AMS radiocarbon method at the Beta Analytic laboratory, using a sample of fauna (from a medium-size ungulate) taken from Sublevel OL.2b of the inner excavation. The conventional dating result was originally calibrated and published using the INTCAL13 calibration curve [3,34], and it has now been re-calibrated with INTCAL20 [35].

## 3. Geomorphology

El Olivo Cave is a limestone conduit with a rounded ceiling of phreatic/epiphreatic origin (Figure 4A), suggesting the formation of the cave passage at a paleo-water table. Cave walls exhibit flowstone preserved as ledges perched 2 m above the detrital infill (cave floor) containing the archaeomaterials (Figure 4B–D). The flowstone would have precipitated on a detrital deposit, partially or totally removed later by natural erosion. Therefore, the perched flowstone marks the occurrence of an earlier cave infill (represented as the former cave floor in Figure 4C). A sample has been taken from this flowstone for dating using the U/Th method. Afterward, a period of fluvial incision occurred because El Olivo Cave is 13 m higher than the Cabornio stream belonging to the Aboño River system (Figure 4C). The present cave infill is detailed in Section 4 and includes fluvial sediments (Figure 4E), breakdown deposits, and archaeological remains.

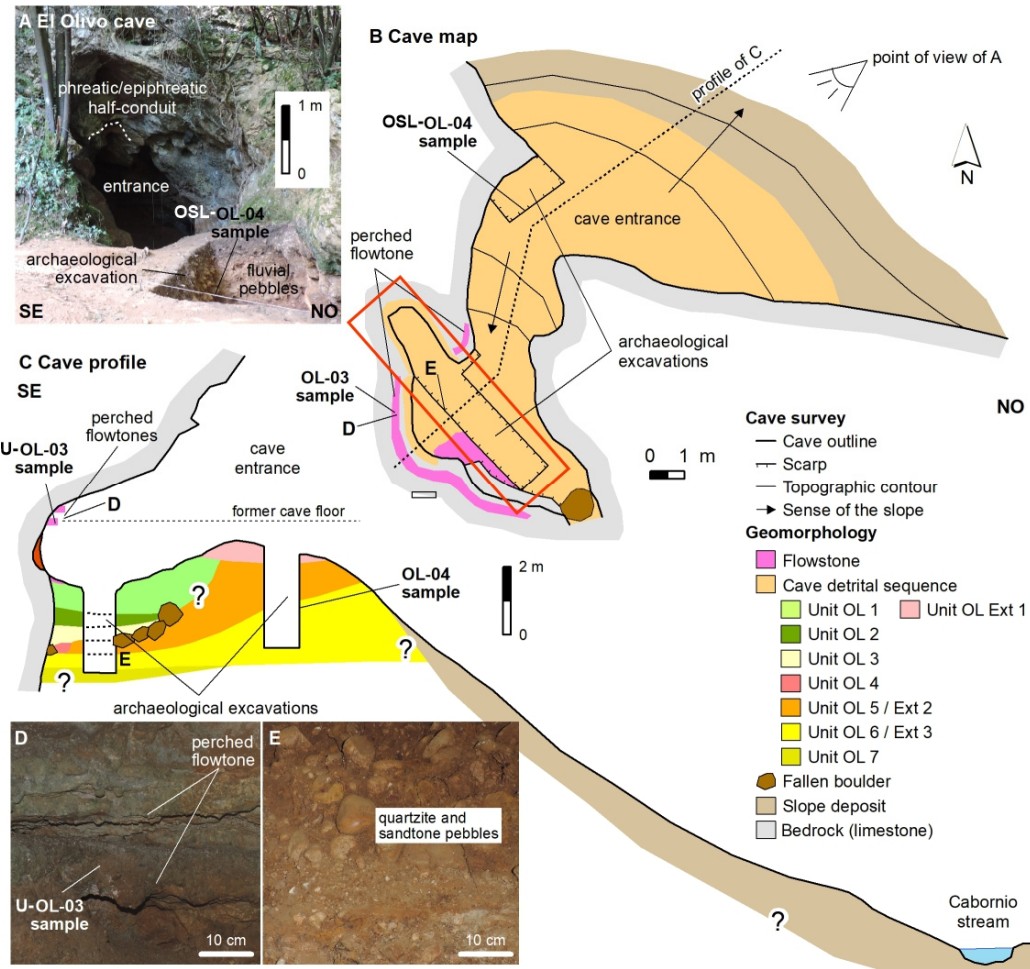

**Figure 4.** (**A**) El Olivo Cave showing the archaeological excavation into fluvial deposits at the cave entrance. (**B**) The cave geomorphological map with the perched flowstone projected outside the cave outline is preserved on the cave walls. (**C**) Profile from the cave to the present-day fluvial network, showing the position of the perched flowstone (which marks a former cave floor) and the cave detrital

infill made of fluvial and breakdown sediments with archaeological remains. (**D**) Perched flowstone and sample. (**E**) Fluvial deposits with quartzite and sandstone pebbles reported in the archaeological excavation performed inside the cave. Red rectangle: area of the photographs in Figure 6.

The Aboño River once flowed through a fluvial basin covering an area of 123 km$^2$, extending northward towards the Cantabrian Sea, which lies just 15 km away. The river basin shows narrow V-shaped valleys resulting from fluvial incisions during the Quaternary. Surrounding the cave under study, the fluvial basin showcases alluvial, karst, slope, and anthropic deposits depicted in Figure 5. The alluvial deposits containing quartzite and sandstone pebbles found in the cave and the Aboño River basin are likely derived from the erosion of Paleogene siliceous conglomerate and sandstone located approximately 530 m S of El Olivo Cave (Figure 5). These detrital rocks are situated on the Cretaceous bedrock and form the Llanera plain, an approximately 100 km$^2$ paleosurface that rises to 200–250 m altitude. The steep slope of the headwaters of the Aboño River indicates the migration of the water divides to the S, resulting in the Aboño watershed capturing the Llanera plain.

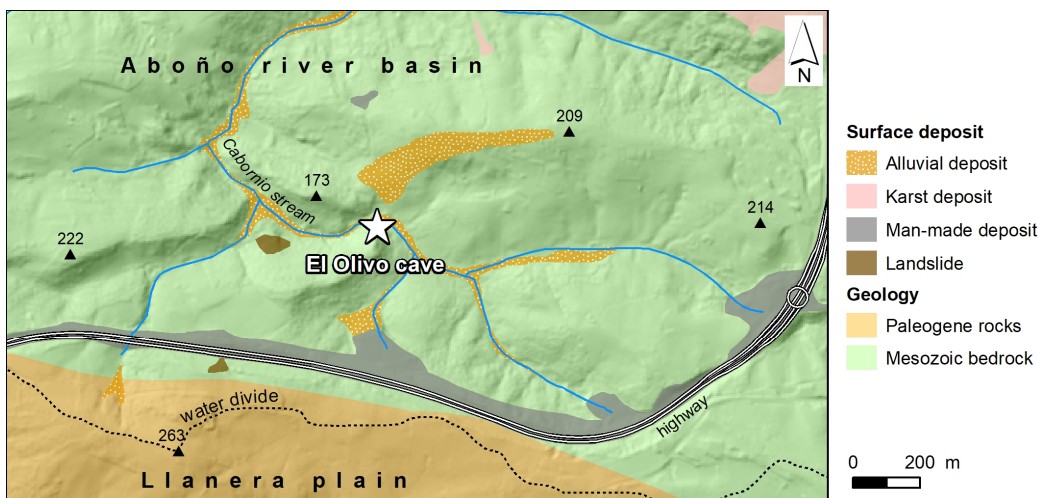

**Figure 5.** Alluvial deposits associated with the Cabornio stream in the surroundings of El Olivo Cave. Alluvial sediments came from the erosion of the Llanera plain, made of Paleogene conglomerate and sandstone. The plain surface is being captured by the Aboño River basin at the present time. The star indicates the location of the El Olivo Cave. Numbers indicate the altitudes above sea level in m of the main elevations.

## 4. Geoarchaeology
### 4.1. The Sequence
#### 4.1.1. Lithostratigraphy

El Olivo Cave is partially filled, although its sediments reached higher levels in the past. Proof of this is visible on the walls of the cave, where remains of speleothems can be observed 2 m above the current floor, where it was sampled for U/Th dating. At the same time, numerous patches of sand are adhered to the walls below the speleothems as remnants of a first fill that was dismantled. To the right, the cave continues in an N-NW direction through a narrow passage or sink, which was also filled before the excavations began. The archaeological work in the cave allowed us to observe that the sink continues inwards through a conduit, which was also filled with clastic sediments of autochthonous character. In this area, water circulation followed the wall–ceiling plane of the drain, which caused a dragging of sediments toward the interior of the cave. The sink was filled with autochthonous boulders that partially blocked it. These boulders were embedded in fine orange-colored sediments on which a package of fluvial boulders rested in a highly erosive manner. In the innermost part, the accumulation of these sediments and rocks completely

filled the channel. Above all these deposits, in the cave's main chamber, other more recent deposits with a partially mixed appearance were deposited.

In the archaeological trench excavated inside the cave, the following levels are observed from bottom to top (Figures 6 and 7):

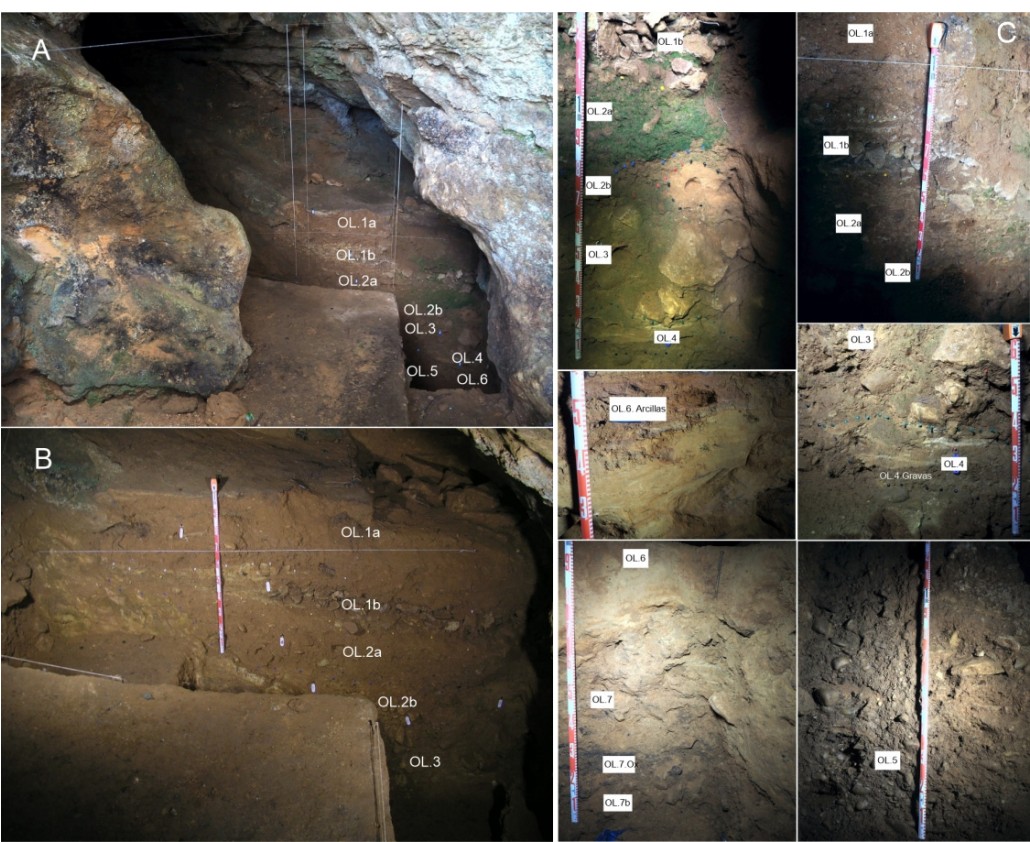

**Figure 6.** El Olivo Cave. (**A**) General view of the stratigraphic profile from the entrance. (**B**) Detail of the previous view. (**C**) Details of the stratigraphic sequence.

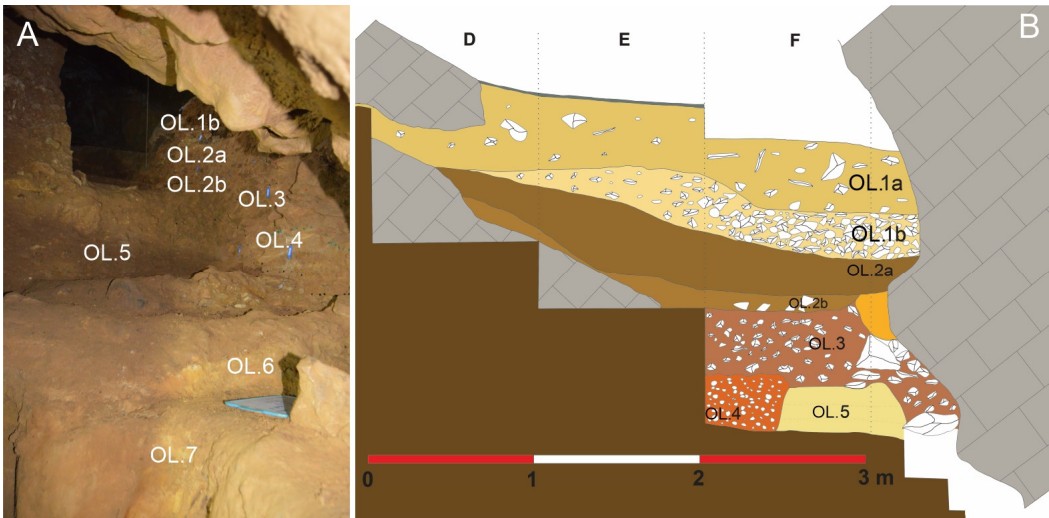

**Figure 7.** El Olivo Cave. Interior trench stratigraphy: (**A**) photography and (**B**) stratigraphic section. In (**B**), capital letters D, E and F correspond to the name of the squares of the excavation.

**OL.7 Level:** Known basal level of the sequence. More than 60 cm of reddish-brown clays and black spots due to dark oxides, with fine and medium quartz sands, well rounded

and abundant gravel and pebbles (10 cm centile and 3 cm average) arranged chaotically. Most of them are autochthonous limestone clasts with very irregular geometry and sharp edges, together with other moderately rounded limestone clasts. It is structured in two sections: the lowermost (sample OL.7b), characterized by the greater presence of dark oxides, especially at the top (sample OL.7 Ox) that stain the faces of the limestone clasts; and the uppermost (sample OL.7a), with less presence of oxides, separated from the previous one by a horizontal thin level of dark oxides. It contains interspersed lithic and faunal remains arranged chaotically in the deposit. The lithic industry is no longer laminar, quartzite is more abundant than flint, and the fauna is large. The two samples sent for radiocarbon dating (AMS direct and Beta Analytic) lacked sufficient collagen because the bones were highly altered. There is plentiful evidence of dark oxides on the faunal remains, and it is possible that the level was formed in a context with abundant water. However, the recovered lithic collection is very scarce [4,5]. The base of this level has not been reached.

**OL.6 Level:** Contains 40 cm of well-rounded fine and medium quartz sand, predominantly yellow with silt and clay (sample OL.6), which appears finely laminated with a depositional slope towards the cave's interior (N-NW). The sand is affected by hydromorphism that results in a mottled coloration (white, yellow, and orange). This level is strongly erosive on OL.7, and planar carbonate concretions appear in some areas in the contact between both. Ascending up the passage, it is possible to observe how the sands vertically pass to some red clay with silt and very fine quartz sand and some gravel (sample OL.6. Arcillas), massive, inclined towards the interior with a carbonated crust on top and a desiccation crack filled by carbonates. It is heavily eroded by OL5 and OL.4. It presents faunal remains that show the action of carnivores at the top of the level.

**OL.5 Level:** From 30 to 120 cm of a heterometric clast-supported conglomerate (sample OL.5), yellowish with gray tones, formed by quartzite pebbles and quartz gravels, spherical to subspherical and very well rounded (12 cm centile and mean 2 cm), with a scarce reddish matrix of clay and silt with coarse, fine and very fine quartz sand. The fine fraction is composed mainly of quartz grains with little presence of calcite. The size of the clasts varies from 2 to 12 cm from bottom to top, resulting in a grain-increasing arrangement, although internally, the deposit presents a massive appearance, with a certain horizontal organization of the planar clasts. The uppermost part is heavily eroded, hence its thickness variation, and the erosive scar generated is filled by OL.3, which, as we will see, is made up of angular autochthonous limestone clasts. It contains some isolated quartzite flakes with a strong patina. It can be correlated with the OL.Exterior.2 level of the exterior trench, although the contact between the OL.5 pebbles and the OL.6 sand inside the cave is about 2 m below the same contact in the exterior trench.

**OL.4 Level:** From 20 to 30 cm of clayey silt with fine and medium quartz sand, slightly angular and rounded, ranging in color from white, yellow, orange, and red (sample OL.4), which towards the lowermost part becomes silty-clayey sand with limestone gravel in thin levels (sample OL.4. Gravas). They are laminated with a depositional inclination towards the cave's interior, overlying the large scar that erodes the underlying conglomerates, and they gradually decrease in thickness in a stepped manner until they disappear.

**OL.3 Level:** From 40 to 50 cm of clayey fine sand, light brown with whitish areas (sample OL.3), with scattered gravel and large irregular blocks of autochthonous limestone, both rounded and angular, together with fragments of stalagmitic crusts arranged chaotically towards the interior of the cave. It is partially cemented by carbonates, giving it a breccia-like appearance. The level is disturbed, with isolated faunal and lithic remains.

**OL.2 Level**: Deposit of yellow sand internally structured in two sections:

**OL.2b:** Lower section, 0 to 10 cm thick, formed by yellow quartz sand, fine and very fine, well rounded, silty and clayey, with whitish tones and some angular limestone pebbles and gravel (2 cm centile) (sample OL.2b). The fine fraction is predominantly quartz with very little presence of sodium feldspar and calcite. It wedges towards the east, resting on the underlying rock of the substratum, while towards the west, it is abruptly interrupted without reaching the wall, as it is supported by a patch of yellow sand attached to the wall,

possibly a remnant of a previously eroded deposit. Its appearance is massive. It does not appear to be altered. A fauna sample from this sub-level has provided a radiocarbon age of 13,960 ± 40 BP [4].

**OL.2a:** Upper section, 35 to 40 cm of yellow quartz sand, fine to very fine, well rounded, clayey and silty, with scattered gravel and pebbles (sample OL.2a). It includes angular limestone pebbles (centile 7 cm and mean 2 cm) and some scattered red sandstone pebbles, as well as small-sized rounded pebbles (centile 3 cm) and well-rounded quartzite gravels, arranged chaotically. The fine fraction (silt and clay) is mainly composed of quartz with a limited presence of calcite. It is wedged laterally, and its overall appearance is massive. It contains soft mud pebbles up to 8 cm in diameter, fragments of carbonized organic matter, and archaeological remains from the Middle Magdalenian period [9]. The upper part shows traces of alteration due to the intrusion of isolated modern material.

**OL.1 Level:** Reworked deposit with modern materials (ceramics, earthenware, bullet casings, and glass) and Palaeolithic artifacts (lithic and fauna from Level OL.2) that can be divided into two sections:

**OL.1b:** Lower section 25 cm thick of clast-supported conglomerate with autochthonous limestone clasts and rounded pebbles with a limited clayey-silty matrix. Laterally, it transitions to yellow sands and silts. It wedges laterally and exhibits strong erosive characteristics on the underlying level.

**OL.1a:** Upper section formed by 45 cm of reddish-brown to orangish sands and silts, with some autochthonous clasts and pebbles.

The rounded pebbles in this level appear to originate from the dismantling or excavation of level OL.Exterior.2 in the part of the entrance that occurred in modern times, creating this artificial stratigraphy.

**Surface level:** A 2 cm thick dark layer composed of organic matter, possibly resulting from the modern use of the cave. It is highly compacted due to trampling.

Attached to the sink's left wall (W) is a small patch 30 cm wide and 20 cm thick, composed of medium, fine, and very fine, well-rounded quartz sand. It is likely a remnant from a previous fill (**OL.Arenas anteriores**) (Figure 8).

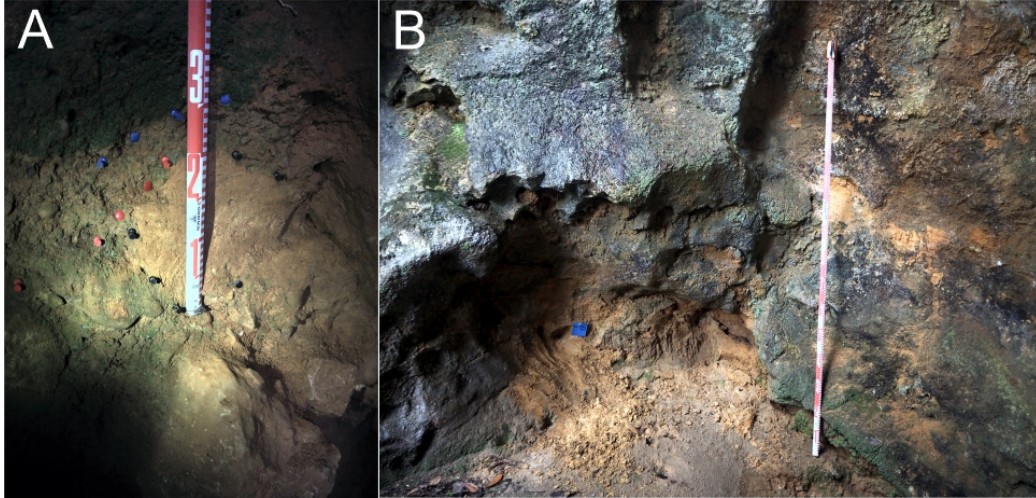

**Figure 8.** El Olivo Cave. Photographs of the sampling points of OL.Arenas anteriores (**A**) and OL.Arenas exteriors (**B**).

We also took a sample of the orange sand outside the cave (**OL.Arenas exteriores**), which adhered to the wall. It consists of medium quartz sand grains that are well rounded (Figure 8).

In the archaeological excavation conducted outside the cave (Squares G1 and G2), on the existing terrace in the access area to the cave, a sequence can be observed consisting of, from bottom to top (Figure 9):

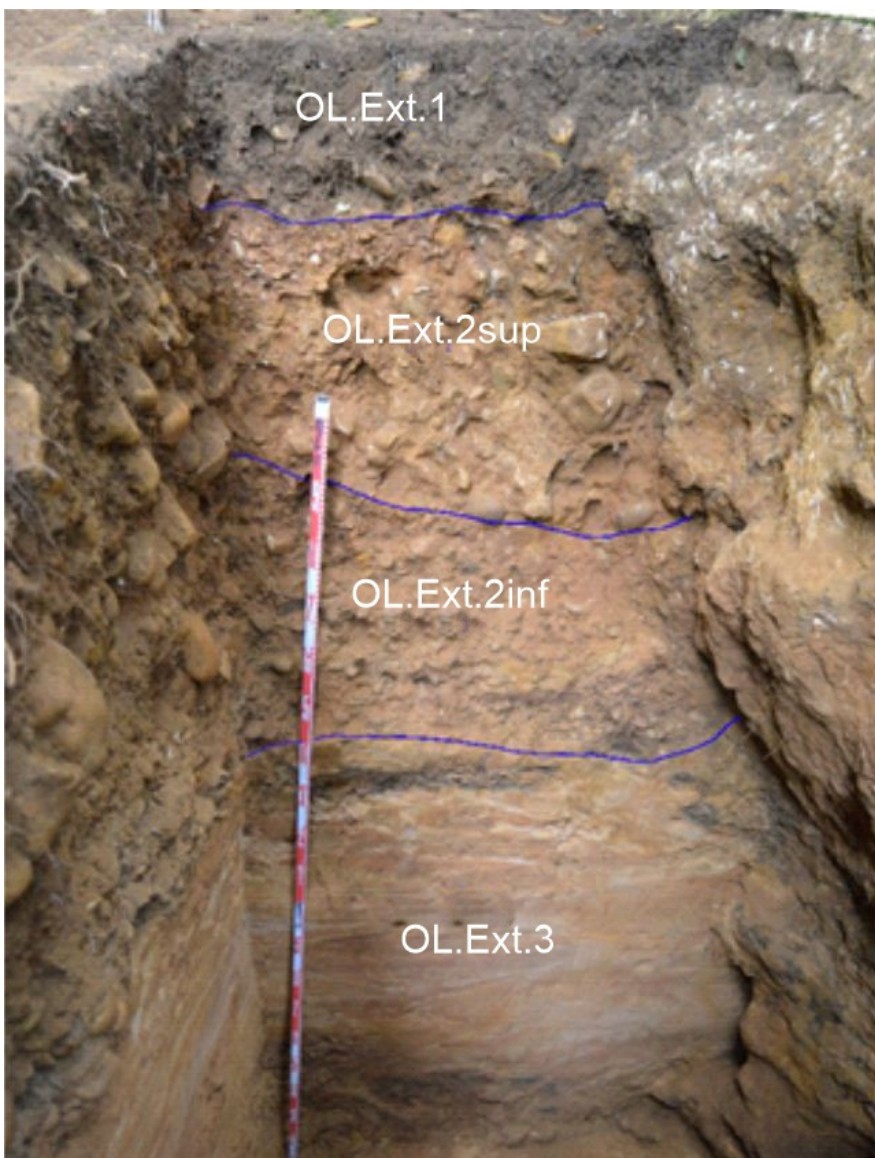

**Figure 9.** El Olivo Cave. Stratigraphy outside the cave (the scale is 2 m).

**OL. Exterior.3 Level:** (140 cm visible) Medium, fine, and very fine quartz sand finely laminated horizontally, light orange in color, similar to the sand in the interior level OL.6. A sample taken from the uppermost part of this level for OSL dating provided a date of 23,500 ± 6200 years old (we use the expression "years old" because, in the case of OSL dating, it is not appropriate to use the term "BP", which should be restricted to radiocarbon dates, as was recently pointed out [36]).

**OL. Exterior.2 Level:** (95 cm) Heterometric and growing grain conglomerate, with a clast-supported structure, containing a matrix of medium and fine quartz sands, well rounded, reddish-brown in color, similar to the interior level OL.5. It is divided into two sections: a lower section with fewer pebbles and gravel (IIb), and an upper section with abundant rounded clasts (IIa)

**OL. Exterior.1 Level:** (40 cm) Dark brown to black clay, with quartz sand and abundant organic matter.

### 4.1.2. Archaeology

In level OL.7, although the lithic assemblage recovered is very scarce, it presents characteristics that allow us to assign this level to the Middle Palaeolithic provisionally. Laminar production is absent, with a predominance of sidescrapers and denticulated.

Levels OL.6 and OL.5 are practically sterile from an archaeological point of view, with 3 and 7 lithic pieces, respectively, which are not diagnostic.

Level OL.4 contains 122 lithic remains, where laminar supports predominate and a basal fragment of a single basal bevel sagaie stands out. Due to the type of pieces it contains and its chronostratigraphic position in the sequence, its probable chronocultural ascription is to the Lower Magdalenian.

Only 15 pieces of flint lithics were recovered from level OL.3, with a predominance of laminar production. Given the characteristics of this level, it is a reworked context with secondary position materials that probably come from the underlying level OL.4.

Level OL.2, with a total number of 74 pieces, including burins, endscrapers, and retouched blades, as well as a decorated sagaie, fits well in the context of the Middle Magdalenian, both for the lithic and bone industry, as well as for the AMS dating obtained.

### 4.2. Sedimentological and Edaphological Analysis

### 4.2.1. Granulometry

The granulometric analyses of samples from the excavation inside El Olivo Cave have identified a fining-upward detrital sequence, with a predominance of the fine fraction (sand, silt, and clay), with three pulses of gravel and pebble input in the base, middle, and upper part, with the latter being less intense. Sand is predominant in the fine fraction, with a higher presence in the basal and middle parts, while silt and clay increase slightly towards the top (Figure 10).

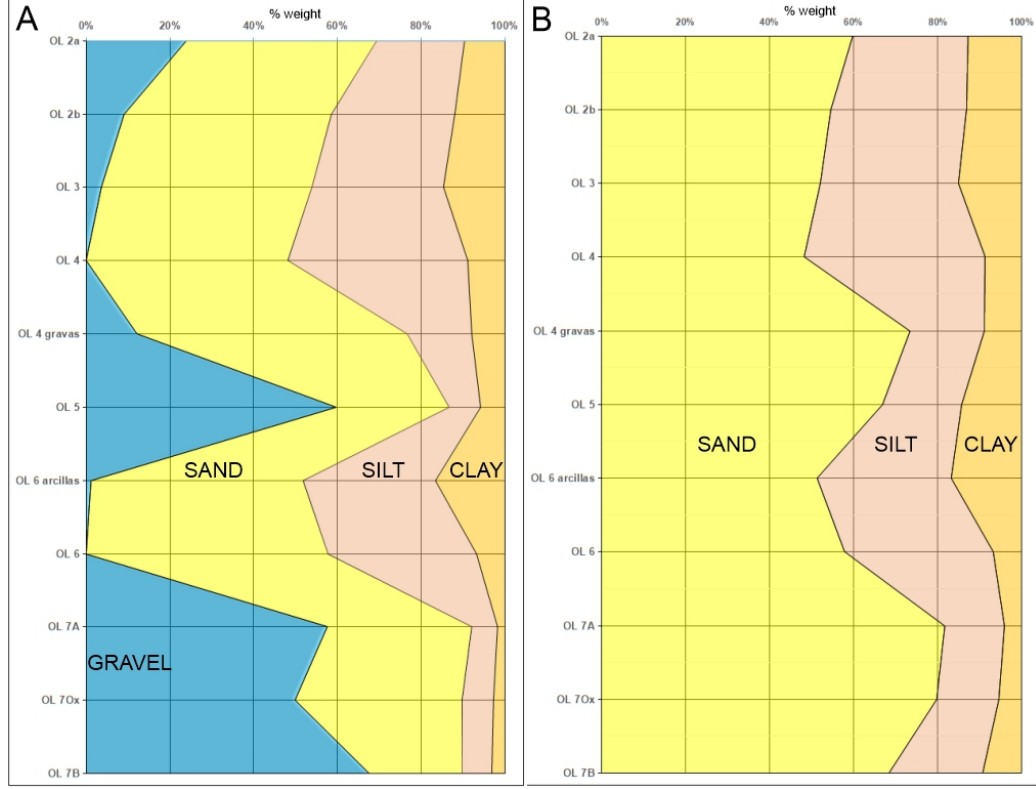

**Figure 10.** El Olivo Cave, inner excavation. (**A**) Granulometry of the total fraction. (**B**) Granulometry of the fine fraction (<2 mm).

In the outer excavation, sand predominates in its lower section, whereas pebbles and gravel dominate in its middle section. Silt and clay are prevalent in the upper section (Figure 11). In the samples obtained from the interior and exterior walls of the cave, sand predominates with a low proportion of silt and clay (Figure 11).

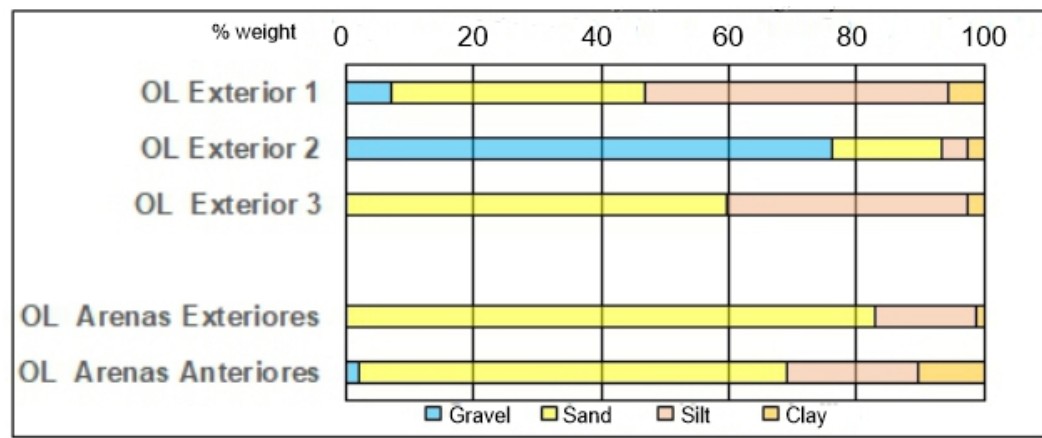

**Figure 11.** El Olivo Cave, outer excavation and samples of sand; granulometry of the total fraction.

As observed in the triangular diagram representing the overall grain size distribution (gravel, sand, and mud) of the samples (Figure 12A), there is a certain homogeneity in the sediments forming the levels in El Olivo Cave. They fall within the range of silty, muddy sands with varying amounts of gravel and sandy gravel with silt and clay. This allows for the differentiation of two sediment groups with slightly different meanings:

Group A encompasses sediments from the textural group of muddy sandy gravel (OL.5, OL.7a, OL.7b, OL.7 Ox, and OL.Exterior.2);

Group B consists of sediments corresponding to the textural groups of gravelly muddy sand (OL.2a, OL.2b, and OL.4 gravas), slightly gravelly muddy sand (OL3, Ol.6 Arcillas, and OL.Arenas anteriores), muddy sand (OL.6, OL.Exterior 3 and OL.Arenas Exteriores), gravelly mud (OL.Exterior 1), and sandy mud.

If we exclude the gravel fraction and focus on grain sizes smaller than 2 mm (sand, silt, and clay) (Figure 12B), the sediment homogeneity of the different units increases, as most of the samples fall within the group of silty sand, except for two samples that correspond to sandy silt (OL.4 and OL.Exterior 1) and muddy sand.

Both diagrams exhibit very similar characteristics to those described for the fluvial sediments of Coímbre Cave (Peñamellera Alta, Asturias) [37].

The granulometric curves of the overall fraction are also quite homogeneous, but four families can be distinguished within them (Figure 13):

- G-A Family: Includes the samples belonging to Group A in the triangular diagram of the total fraction (OL.5, OL.7a, OL.7b, OL.7 Ox, and OL.Exterior 2), which exhibit curves with an initial segment dominated by fine gravel and very coarse to fine sand, accounting for approximately 80 to 90% of the sediment. This is followed by a flatter segment containing very fine sands, silts, and clays, which comprise around 20% of the sediment (Figure 13). It corresponds to two types of deposits: on the one hand, clast-supported conglomerates with a limited matrix, indicating high-energy environments with subsequent settling of the finer particles that make up the matrix (OL.5 and OL.Exterior 2); and on the other hand, debris flow deposits with a minimal matrix that include both fluvial-derived and autochthonous clasts;

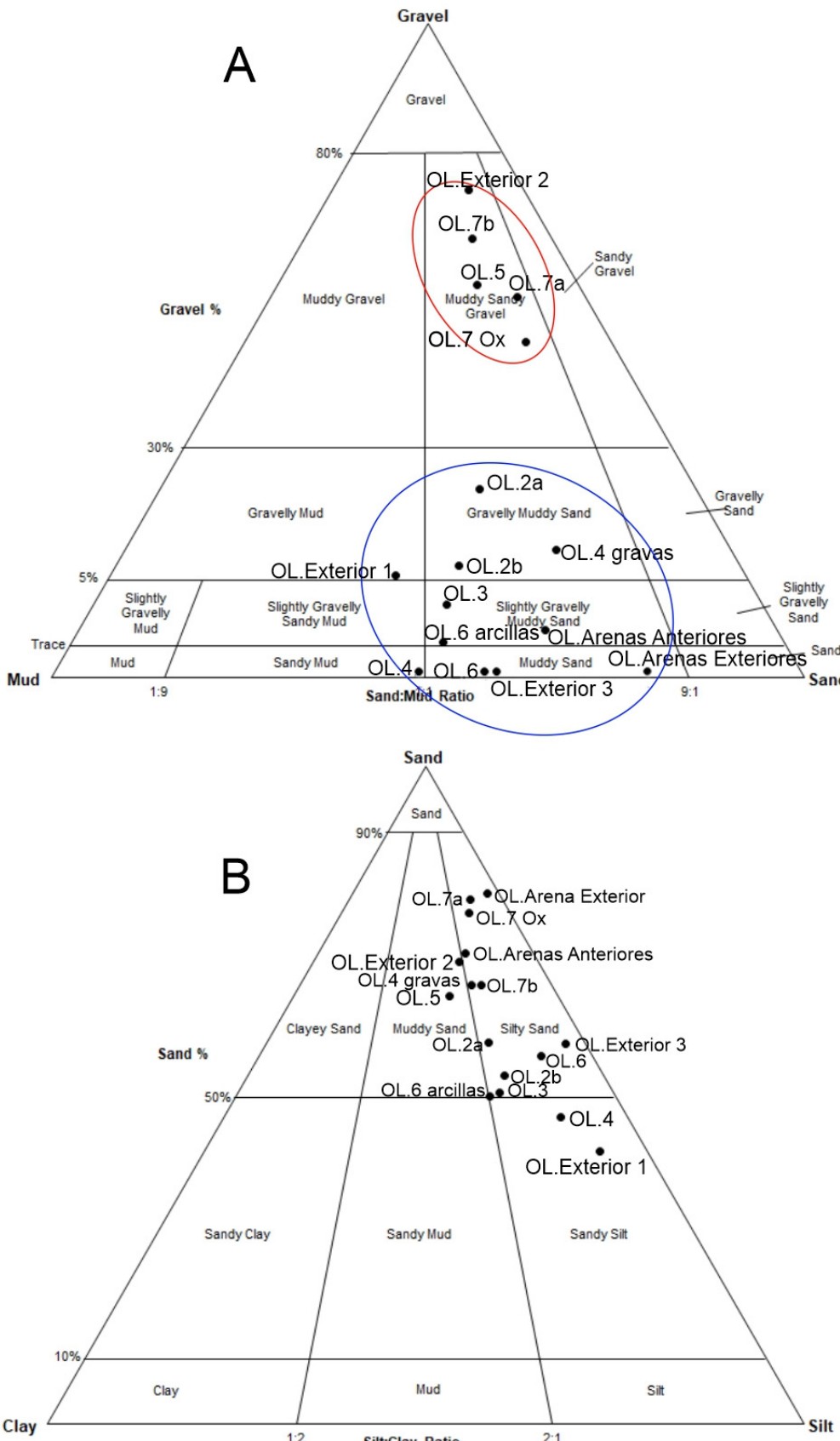

**Figure 12.** El Olivo Cave. (**A**) A triangular diagram representing the total fraction of the sediments from the interior and exterior excavations. (**B**) A triangular diagram representing the fine fraction (<2 mm) of sediments from the interior and exterior excavations.

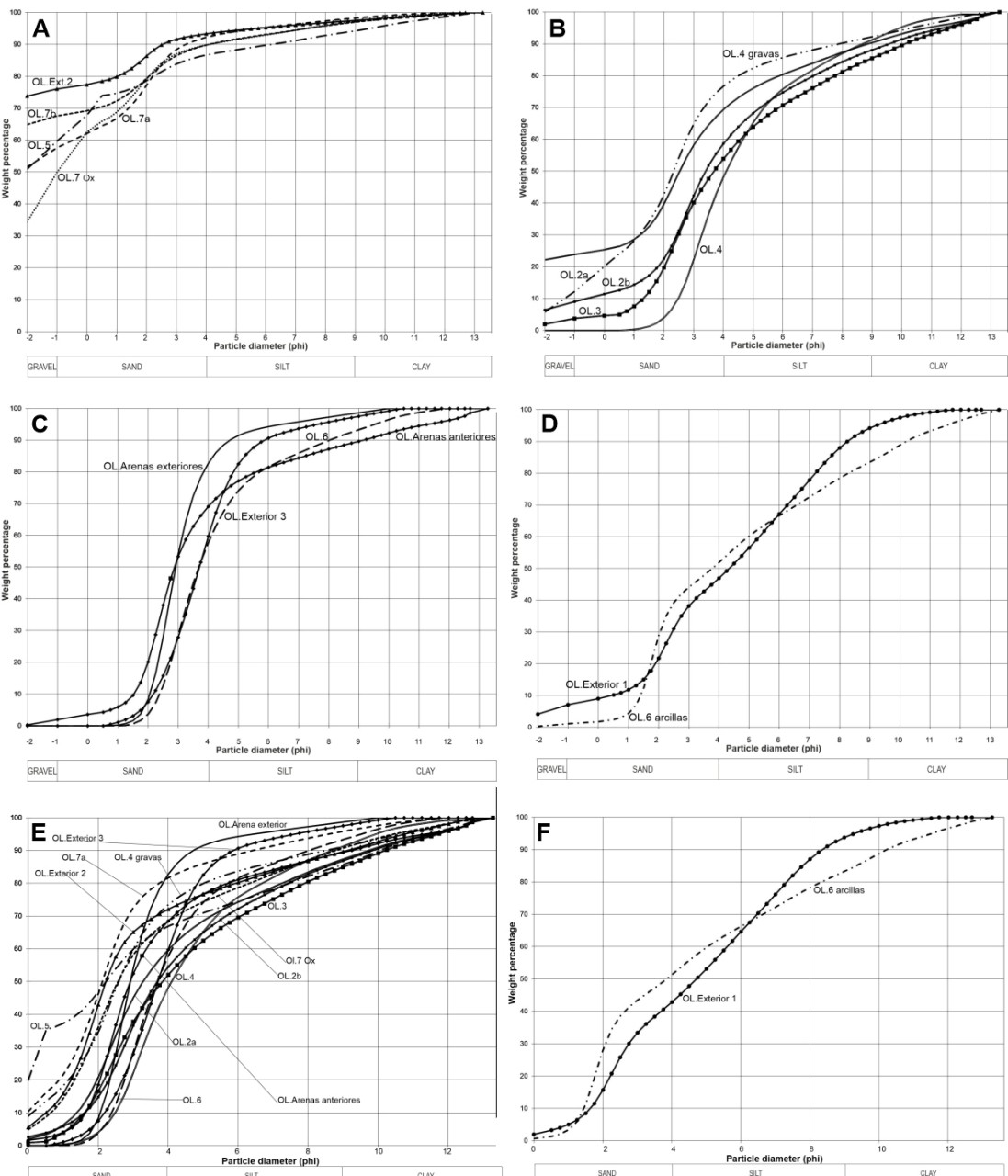

**Figure 13.** El Olivo Cave. Granulometric curves representing the total fraction of the samples of G-A (**A**), G-B1 (**B**), G-B2 (**C**), and G-B3 (**D**) families, and the fine fraction of the samples of F-1 (**E**) and F-2 (**F**) families.

G-B1 Family: It includes the samples OL.2a, OL.2b, OL.3, OL.4, and OL.4 gravas, which exhibit sigmoidal curves with three well-defined segments: an initial flat segment with varying presence of coarse and medium-grained sands, a steep middle segment with abundant fine sands and coarse silts, and a flat final segment with the remaining silts and clays (Figure 13). These curves indicate an essential population centered around fine sand and coarse silt, accompanied by silts, clays, and varying amounts of coarse sand and gravel. They indicate a typically fluvial environment with high to medium energy, characterized by freight transport through reptation, saltation, and suspension;

G-B2 family: It includes samples OL.6, OL.Exterior 3, OL.Arenas Anteriores, and OL.Arenas Exteriores that exhibit curves with a strongly sigmoidal shape with three distinct sections. The first section is relatively flat and includes fine gravel and very coarse, coarse, and

medium sands. The second section is steep and rapidly ascending, ranging from fine sands to coarse silts. The third section is again relatively flat and consists of the remaining silts and clays, extending to clays (Figure 13). These sections indicate the presence of a dominant population, the central one composed of fine sands and very coarse silts transported by saltation and suspension. These curves are typical of high-energy fluvial environments with a high sorting capacity;

G-B3 family: It includes the samples OL.Exterior 1 and OL.6 Arcillas, which exhibit slightly sigmoidal curves. The first section is relatively flat and includes fine gravel and very coarse sand. The second section is steep and corresponds to an increase in the remaining sands until it reaches 50% of the sample. The final section, consisting of silt and clay (Figure 13), represents the remaining portion of the sample. This corresponds to fluvial sedimentation, where significant settling follows the initial bedload deposition.

In the curves of the fine fraction (<2 mm), two families can be distinguished:

F-1 family: It includes the curves of the samples from G-A, G-B1, and G-B2 (Figure 13). The curves of this family present a typically fluvial morphology, as occurs in the total granulometry with the curves of the G.B1, GB.2, and GB.3 families;

F-2 family: It is identical to the G-B3 family of the coarse fraction (OL.6 Arcillas, OL.Exterior.1) (Figure 13). They are fluvial curves where important decantation follows the initial bedload deposition.

The curves of the samples that make up the families of the total fraction G-A, G-B1, and G-B2, in the case of the fine fraction, are unified, given that by not counting the clasts larger than 2 mm, the curves of the fine sediments (sands, silts, and clays) are all very similar, and can be grouped into a single family (F-1). The F-2 family curves are still very different morphologically from those of F-1, as is the case with the total fraction curves.

In general, the deposits of the archaeo-sedimentary sequence of the El Olivo Cave correspond to fluvial sedimentation. The morphological characteristics of the curves in the described families, both in the overall and fine fraction, show significant similarities with the grain size curves of predominantly sandy fluvial sediments from Coímbre Cave [37].

4.2.2. Mineralogy

The interior and exterior excavation sequences are homogeneous in their mineralogy (Figure 14). They are characterized by the predominance of quartz, accounting for over 80% in all samples except for OL.6 Arcillas, which only reaches 70%. Phyllosilicates are present in all samples, but their percentages are below 15%, except for OL.6 Arcillas, OL.4, and OL.3. There is a slight increase in OL.2b (3%) and OL.2a (4%). In the exterior sequence, calcite is only found in the middle section, specifically in OL.Exterior 2 (3%). Goethite is present throughout the interior sequence and in the middle and upper levels of the exterior sequence but with values below 8%.

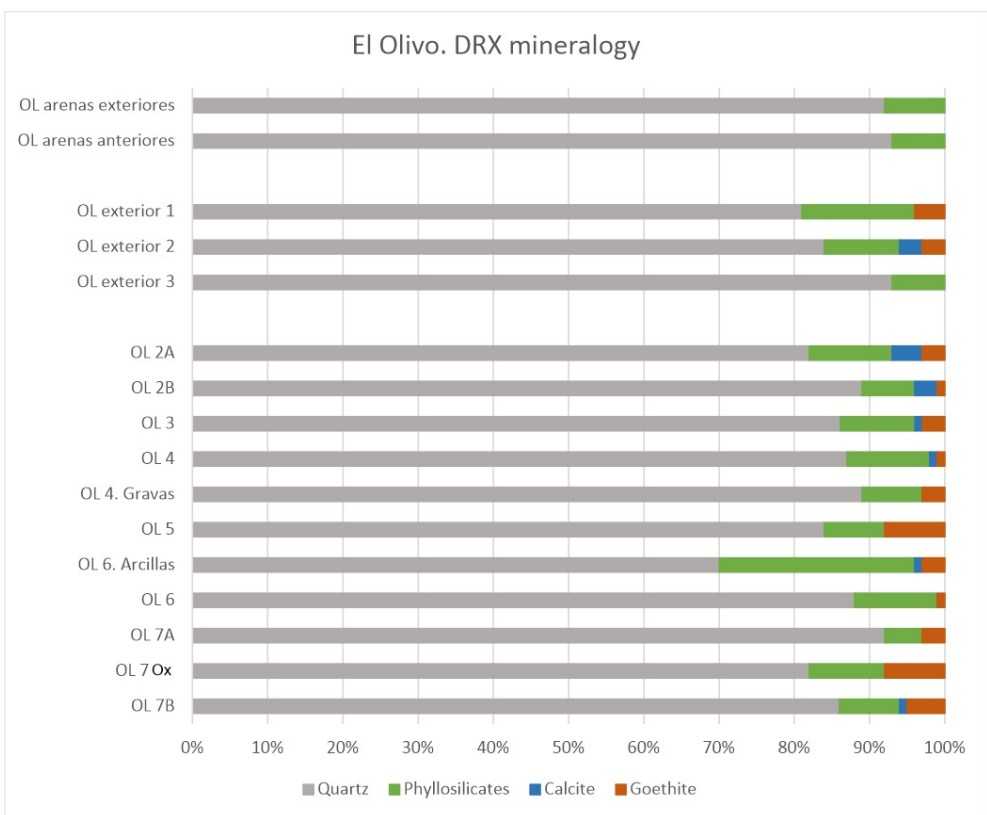

**Figure 14.** El Olivo Cave. XRD mineralogy of the fine fraction of the sediments from the interior and exterior excavations.

The minerals that appear in the different levels (quartz, phyllosilicates, calcite, and goethite) are frequently found in detrital deposits of karstic cavities [38]. The presence of silicate minerals in the fine fraction of the sediments is compatible with the mineralogy of the rocks in the areas surrounding the cavity. It corresponds to tectosilicates and phyllosilicates of allochthonous origin since the limestone in which the cavity develops is very pure. The levels of the interior sequence's upper section and the exterior's intermediate level contain autochthonous calcite, related to the presence of limestone clasts in those same levels.

### 4.2.3. Calcium Carbonate, Organic Charcoal, and Organic Matter

In the interior excavation, the presence of these three components is minimal (<1%), with variations in $CaCO_3$ that notably increase in the samples OL.6 Arcillas (2.68%), OL.2b (4.2%), and OL.2a (2.04%) (Figure 15). The presence of organic matter (OM) and organic charcoal (OC) also increases in the OL.4 sample, reaching 3.47% and 2.01%, respectively (Figure 15). In the exterior excavation, the basal level lacks $CaCO_3$ and has a minimal presence of OM and OC. The middle level exhibits almost 5% $CaCO_3$ and minimal levels of OM and OC, which experience a significant increase in the upper level while $CaCO_3$ disappears. The samples labeled "arenas anteriores" and "arenas exteriores" hardly contain these components.

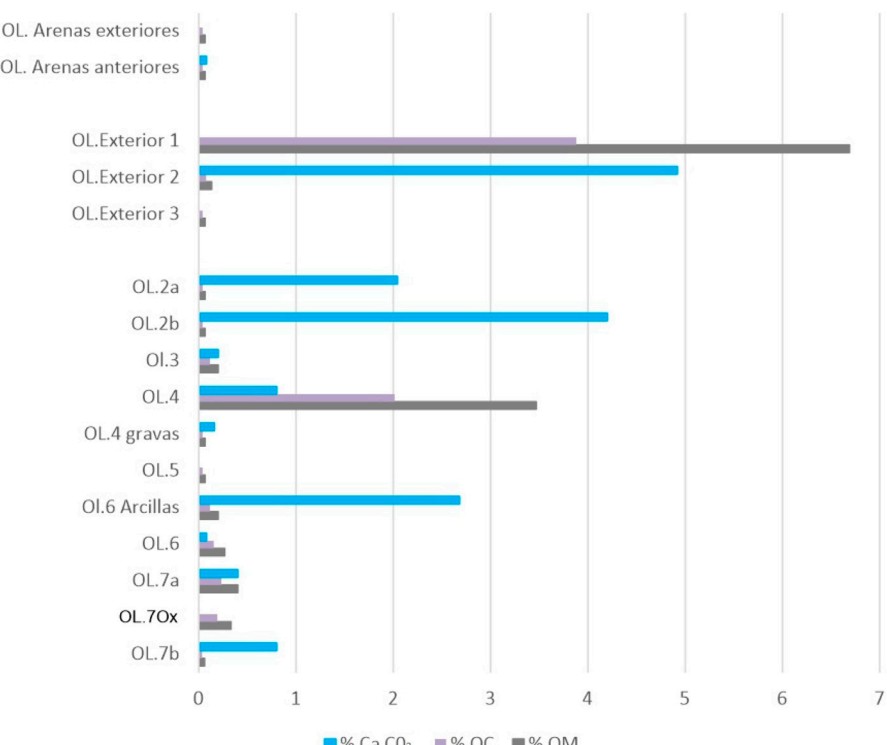

**Figure 15.** El Olivo Cave. Calcium carbonate, organic carbon, and organic matter of the sediments from the interior and exterior excavations.

The highest percentage of $CaCO_3$ is related to the presence of limestone clasts, while the highest presence of organic matter and organic carbon is related to human presence (OL4) and edaphic processes (OL.Exterior 2).

### 4.2.4. Color and pH

From determining the dry color of the sediments in the exterior sequence, certain differences can be observed between the colors of its different levels. The basal level OL.7 exhibits a brown color with a hue of 10YR and 7.5YR and high brightness and chroma values. On the other hand, OL.6 is beige with high brightness and chroma values. The middle section, between OL.5 and OL.4, displays brown colors with a hue of 7.5 YR, ranging from darker tones with low brightness and chroma to darker tones with higher brightness and chroma. The OL.3 level exhibits a reddish-brown color with a hue of 5YR and high brightness and chroma values. OL.2b, a yellowish-brown color, returns to a hue of 10YR with high values of brightness and chroma, while OL2a, a brown color, has a hue of 7.5YR with high brightness and chroma values. The wet color follows a similar pattern with greater color homogeneity. In the exterior sequence, there is a progressive darkening of the color, with a hue of 10YR and brightness and chroma values transitioning from high to low in the uppermost part, both in dry and wet conditions. The "arenas anteriores" and "arenas exteriors" are brown with a hue of 7.5 YR and identical brightness and chroma in wet conditions.

The dark colors are related to the greater presence of organic matter, either from anthropic contribution (OL.5) or from edaphic origin (OL.Exterior 1), and with the presence of iron oxides at the OL.7 Ox level.

The pH values (Figure 16) of the sediments in the interior excavation sequence are slightly basic, around 8, with a minimum in OL.7 Ox. However, there is a significant variation in pH in the exterior core sequence. The basal level exhibits a pH close to 7, which increases to 8 in the middle level and then becomes slightly acidic (6.91) in the uppermost part. The "arenas anteriores" and "arenas exteriores" samples have the highest basic pH values among the samples (8.25).

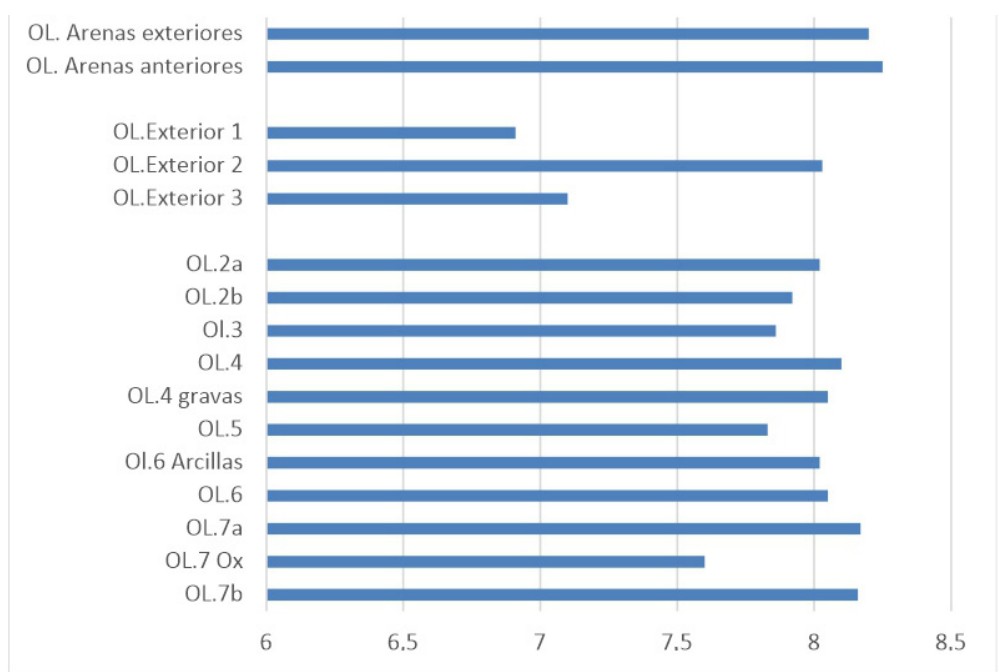

**Figure 16.** El Olivo Cave. pH of the sediments from the interior and exterior excavations.

## 5. Geochronology

The chronological framework of the archaeological site (Table 1) is derived from three U/Th, OSL, and radiocarbon ages. The perched flowstone sample U-OL-03 has a $^{234}U/^{238}U$ isotopic ratio close to 1, indicating that the geochemical system remained closed after speleothem precipitation. Furthermore, the detrital contamination is negligible due to the $^{230}Th/^{232}Th$ ratio being higher than 2. As a result, the age of 189 ± 17 ka for flowstone OL-03 is robust despite the lower concentration of $^{238}U$ (0.46 ppm). This U/Th age marks the end of the former cave sedimentary aggradation within El Olivo Cave during OIS-7a.

**Table 1.** Dates obtained in El Olivo Cave.

| El Olivo Cave. U/Th date | | | | | | | |
|---|---|---|---|---|---|---|---|
| Unit | Sample | Lab code | $_{238}U$ (ppm) | $_{232}Th$ (ppm) | $_{234}U/_{238}U$ | $_{230}Th/_{232}Th$ | $_{230}Th/_{234}U$ | Date |
| Flowstone | U-OL-3 | 917 | 0.46 | 0.58 | 1.05 ± 0.02 | 2128 ± 0.062 | 0.83 ± 0.03 | 188,927 + 18,323/−15,739 |
| El Olivo Cave. OSL date | | | | | | | |
| Unit | Sample | | beta dose (Gy/ka) | gamma dose (Gy/ka) | cósmic dose (Gy/ka) | equivalent dose (Gy/ka) | annual dose (Gy/ka) | Date (ka) |
| OL.Exterior.3 | OSL-OL-4 | | 0.66 ± 0.17 | 0.046 ± 0.12 | 0.13 ± 0.006 | 29.5 ± 3.5 | 1.25 ± 0.30 | 23.5 ± 6.2 |
| El Olivo Cave. Radiocarbon date | | | | | | | |
| Unit | Sample | Lab code | | Radiocarbon date BP | $_{13}C/_{12}C$ | Calibrated age 2σ cal BP | |
| OL.2b | C14-OL-1 | Beta-375569 | | 13,960 ± 40 | −21 | 17,060–16,830 | |

Sample OSL-OL-04 had a very low radioisotope content (<0.7 Gy·ka$^{-1}$), and no disequilibrium was observed in any U and Th decay chains. The OSL signals were dim but fast. After accepting 35 aliquots, a central age model was applied because the equivalent dose (De) distributions were non-skewed, the over-dispersion was low, and there was no evidence of incomplete bleaching. As a result, the OSL age of 24 ± 6 ka is considered reliable for inferring the timing of the second detrital deposition period, which is coeval with OIS-3a (Table 1).

Finally, Table 1 provides the results of the radiocarbon dating obtained from medium-sized ungulate remains recovered in the sublevel C14-OL.2b of the interior excavation. The

conventional results indicate an age of 13,960 ± 40 BP, corresponding to the calibrated range of 17,096–16,811 cal BP with 95% probability using the INTCAL20 curve. This places the deposit and the remains of the Middle Magdalenian period it contains at the beginning of the cold stage GS 2a of the Last Glacial Maximum within the OIS2 (Oxygen Isotope Stage 2), coinciding with the onset of the Heinrich H1 event at the end of the Upper Pleistocene.

## 6. Geoarchaeological and Geomorphological Interpretation

### 6.1. Geoarchaeological Interpretation

Based on the results of the geomorphological analysis of the surroundings and the cave, as well as the lithostratigraphic description, the chronological data, and the sedimentological and edaphological characterization of the sediments from both the interior and exterior sequences of El Olivo Cave deposits, it appears evident that its genesis is related to typically alluvial sedimentation.

As previously observed, before the river incision, the Paleogene conglomerates and sandstones formed a cover that extended over most of the Cretaceous limestone near the cave. In this context and any case before the Middle Pleistocene, the cave was formed by the circulation of phreatic water flow directed towards the NE. Subsequently, the development and incision of the fluvial network of the Aboño River resulted in the dismantling of the Paleogene cover from the north, while El Olivo Cave was nearly completely filled with fluvial sediments that were sealed by a speleothem dated at 189 ± 17 ka. Evidence of this fluvial deposit includes the sand adhered to the cave's walls in its inner and outer zones, referred to as "arenas anteriores" and "arenas exteriores". These sands are identical and exhibit strong similarities in grain size, mineralogy, $CaCO_3$, OC, and OM with the basal sands of the exterior excavation. However, differences can be observed in terms of pH and color. Despite their geoarchaeological similarity, their different stratigraphic position in the cavity suggests that the two sets of sand do not correspond to the same sedimentary process. The sands adhered to the cave walls are believed to be a part of the initial sedimentation of the cave, which was sealed by the speleothem dated at 189 ± 17 ka. In contrast, the sands in the exterior excavation on the fluvial terrace are more recent. This sedimentary fill was later eroded, leading to the near-complete emptying of the cave; only small remains of sand were preserved in cavities in the walls of the cavity.

When the Cabornio stream had already cut down 70 m into the Paleocene cover, it intersected El Olivo Cave, allowing it to communicate with the topographic surface and resulting in the emptying of the sediments that filled the cave. From that moment on, the cave began to refill with fluvial deposits, with episodes in which prehistoric human groups occupied the cave. Furthermore, during that time, sedimentation of the exterior deposits occurred, forming a small terrace. These detrital deposits, both interior and exterior, originated from the dismantling of the Paleogene cover. Subsequently, the Cabornio stream continued to incise, interrupting the influx of sediments into the cave.

During the Upper Pleistocene, the cave was once again filled with a sequence (interior excavation) that begins with a well-documented sedimentation of muddy conglomerates (level OL.7). These conglomerates comprise both angular autochthonous clasts and fluvial rounded pebbles, with quartz being the predominant mineral accompanied by a low proportion of phyllosilicates. A certain presence of goethite is detected, which stains the surfaces of the clasts black and accumulates in a thin intermediate layer, indicating a process of hydromorphism. These deposits consist of a clast-supported conglomerate that exhibits characteristics of a debris flow. They contain archaeological remains that appear to correspond to the Middle Palaeolithic [4].

The sequence in the exterior trench excavated in this terrace presents a basal level consisting of well-sorted fine sands, primarily composed of quartz with minimal phyllosilicates, no carbonates, and scarce organic matter. These sands exhibit fine horizontal lamination. They were deposited around 23.5 ± 6.2 ka. Above them, a clast-supported and grain-increasing conglomerate consists of rounded pebbles with a matrix of coarse and medium quartz sands, with few fine sands and silts. This conglomerate serves as the substrate for the current soil, which is rich in organic matter and dark in color.

The sequence inside the cave continues with high to medium-energy fluvial sedimentation, contributing deposits transported through reptation, saltation, and suspension. There is an initial low-energy episode where the sedimentation of the sandy bedload occurs, followed by the settling of silts and clays (OL.6). These fluvial sands inside the cave may correlate with the fluvial sands at the base of the exterior terrace (OL.Exterior.3), which are dated around 23.5 ± 6.2 ka, during the final stages of the cold stage OIS 3a (Last Cold Period).

Above these sands, fluvial conglomerate deposits (OL.5) enter the cave and extend into the narrow passage at the NW end, blocking it. These deposits exhibit similar textural and mineralogical characteristics to those in the middle level of the exterior sequence (OL.Exterior 2), although the latter shows a higher presence of $CaCO_3$ compared to the interior levels.

Above these conglomerates, the sequence continues with predominantly sandy fluvial sedimentation. It starts with fluvial gravel (OL.4 gravas), followed by a muddy section (OL.4) associated with a ponding period with a predominance of decantation.

Above that, the sequence continues with fluvial sands containing abundant autochthonous clasts and fragments of speleothems (OL.3), followed by fluvial sand and silt with gravel (OL.2), sedimented during fluvial flood events that contain remains of Middle Magdalenian occupations dated to 17,096–16,811 cal BP [3,4], corresponding to the beginning of the cold stage GS 2a of the Last Glacial Maximum within OIS2.

The landscape during the human occupation of the cave was likely very similar to the present day (with the cave entrance around 10/13 m higher than the Cabornio stream), with well-developed river valleys and gentle hills. Immediately south of the cave would be a small plain corresponding to the current area of Llanera and Noreña. At that time, the distance from the cave to the sea was greater than the current 10 to 15 km, as the sea level was between 115 and 80 m below the current level, which extended the coastal strip between 6 and 15 km [7].

The sequence of the interior excavation concludes with a superficial deposit articulated into two sub-levels. In these reworked fluvial deposits, Palaeolithic archaeological remains are found alongside recent materials. There is evidence of the cave being used during the Spanish Civil War, supported by Mauser bullet casings and the oral accounts of the residents in the area. As far as we know, the cave was used as a refuge, primarily in 1937, during the advance of Francoist troops towards Gijón. Subsequently, while residents are aware of the cave and visit it, there is no record of any more recent activity or use [4]. Outside, the current top soil has developed on the conglomerate of the fluvial terrace.

*6.2. Paleogeographic Evolution*

Based on the geomorphological, geomorphological, and geochronological evidence obtained from El Olivo Cave and its surroundings, we were able to reconstruct the paleogeographic evolution since the Chibanian, as illustrated in Figure 17. The model consists of five phases:

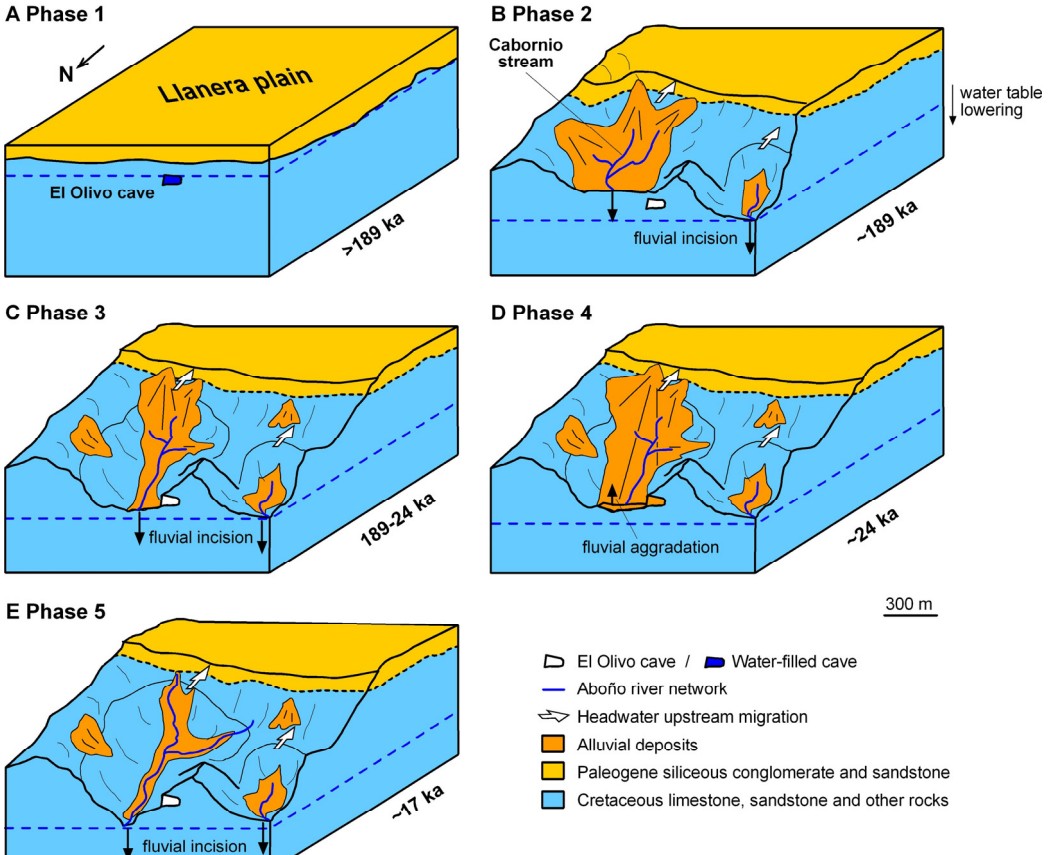

**Figure 17.** Paleogeographic evolution of El Olivo Cave: (**A**) Phase 1: Cave formation within Cretaceous bedrock covered by Paleogene rocks (Llanera plain). (**B**) Phase 2: Erosion of the northern Llanera plain due to fluvial incision and headwater upstream migration; cave aggradation until 189 ± 17 ka. (**C**) Phase 3: Partial erosion of the former cave infill coeval to fluvial incision. (**D**) Phase 4: Alluvial deposition inside the cave in relation to alluvial fans during OIS-2. (**E**) Phase 5: Fluvial incision up to present-day coevally with human frequentation. The blue dashed lines indicate the water table.

The first phase began with the formation of El Olivo Cave within Cretaceous bedrock covered by Paleogene detrital rocks, which formed the Llanera plain (Figure 17A). The cave conduit originated when the water table was located 147 m above the present sea level. Therefore, Phase 1 took place a long time before the precipitation of the flowstone OL-03 at 189 ± 17 ka;

Phase 2 comprised the entrenchment of the Aboño river network on the Llanera plain in the vicinity of El Olivo Cave (Figure 17B). The headwaters migrated southwards, eroding the Llanera plain. The fluvial incision also caused the lowering of the water table and the vadose development of the cave. Finally, the cave was partially filled by detrital sediments and flowstones precipitated at 189 ± 17 ka, coeval with the limit between OIS 7-6. These detrital and speleothem deposits remain perched on the cave walls. The cave infill would be related to the erosion of the Paleogene rocks (Figure 17B) and coincides with a sedimentary aggradation event in karst caves along the Cantabrian Region during OIS 7-6 [39–41];

Fluvial incision, the drop of the water table, and the erosion of the Llanera plain continued during Phase 3 (Figure 17C). At the same time, the cave sedimentary infill was partially removed before or after the interception of the cave by the topographic surface. This led to the creation of the cave entrance, which allowed the potential entrance of fauna and humans, as shown by the probable presence of Neanderthal groups in the cave;

Phase 4 corresponds to the deposition of sandstone, quartzite pebbles, and quartz sand transported by the Cabornio stream from Llanera plain to El Olivo Cave (Figure 17D). This

implies the location of the Cabornio stream channel at the position of the cave. The alluvial deposition within the cave occurred around $24 \pm 6$ ka and would be related to alluvial fans developed under the dry and cold conditions of OIS-2;

Fluvial incision continued during Phase 5 (Figure 17E), and humans frequented El Olivo Cave at the end of OIS-2, according to Álvarez-Alonso et al. (2018) [4]. Simultaneously, the stream flooded the cave, leading to sandy-loamy sediment with reworked archaeological remains during the OIS-2. Cabornio stream has descended 13 m from 24 ka to the present, representing an incision rate of 0.54 mm·a$^{-1}$.

## 7. Conclusions

El Olivo Cave has a long geological history that begins with its formation at an undetermined time in the Neogene/Pleistocene before $189 \pm 17$ ka. During the Lower and Middle Pleistocene, the Aboño River and its tributaries were embedded, and the cave opened to the exterior. It was practically filled by sedimentation of fluvial sands, culminating in a speleothem dated $189 \pm 17$ ka during the end of the warm stage OIS7a in the Chibanian (Middle Pleistocene). Subsequently, these sands were eroded, leaving remnants on the cave walls. Sedimentation resumed inside the cave, resulting in a fluvial clastic sequence. In its lower section, there is a basal level containing archaeological remains tentatively assigned to the Middle Paleolithic, followed by sands that correlate well with a sandy deposit located at the base of the outer terrace, dated to $23.5 \pm 6.2$ ka, at the end of the cold stage OIS3a (Last Cold Period) of the Upper Pleistocene. The fluvial sequence continues, and in its upper section, there is an occupation during the Middle Magdalenian period, dated within the calibrated range of 17,096–16,811 cal BP. The sequence concludes with a disturbed deposit that contains contemporary remains, among which notable artifacts from the Spanish Civil War, including weaponry, can be found.

Examining this archaeological site has yielded novel insights into the evolution of the landscape in central Asturias during the Pleistocene. This region has been constrained by the scarcity of sedimentary records, limiting the development of pertinent Quaternary investigations. The northern central region of Asturias is characterized by continental plains [41] and provides potential habitats for large herbivores, all documented in numerous paleontological and archaeological sites throughout the region [42–44].

The occupation evidence corresponding to the Middle Magdalenian period is situated within the context of significant human presence of a similar chronology in the Nalón Valley. This led to new hypotheses regarding space organization, mobility, and territoriality during the Upper Palaeolithic [3]. Thus, considering a new type of settlement, defined as a "secondary camp" [3] based on the evidence from the OL.2 level, opens the door to new interpretative perspectives regarding the cultural, economic, and social space that defines Magdalenian territories.

**Supplementary Materials:** The following supporting information can be downloaded at https://www.mdpi.com/article/10.3390/geosciences13100301/s1, Table S1: Laser granulometry; Table S2: Granulometry; Table S3: Mineralogy (DRX); Table S4: $CO_3Ca$, OC, OM; Table S5: pH; Tabñe S6: Colour.

**Author Contributions:** J.F.J.P.: conceptualization, field work, methodology, collection of the sample material, writing—original draft, figure preparation, review and editing; D.Á.-A.: conceptualization, field work, methodology, collection of the sample material, writing—original draft, figure preparation, review and editing; M.d.A.-H.: conceptualization, field work, methodology, writing—original draft, review and editing; D.B.: field work, methodology, writing—original draft, figure preparation, and review; P.C., methodology, soil analyses, writing—original draft, figure preparation, and review; A.H.-C.: field work, methodology, collection of the sample material; J.S.: methodology, U/Th analyses, writing—original draft, figure preparation; S.G.: methodology, OSL analyses, writing—original draft, figure preparation; M.J.-S.: field work, methodology, writing—original draft, figure preparation, and review. All authors have read and agreed to the published version of the manuscript.

**Funding:** The excavation in El Olivo Cave was funded by the Llanera City Council and Fluor S.A. This work and this article has been carried out within the projects CantabricOIS2 (PID2020-115192GB-I00), funded by the Government of Spain, and FUO-367-16, funded by the University of Oviedo Foundation. DB is affiliated to Plan Andaluz de Investigación, Desarrollo e Innovación 2020 (Junta de Andalucía, Spain), and both DB and MJS belong to the GEOCANTABRICAE Project (SV-PA-21-AYUD/2021/51766, FEDER-FICYT).

**Institutional Review Board Statement:** Not applicable.

**Data Availability Statement:** Some data are provided in the Supplementary Materials.

**Conflicts of Interest:** The authors declare no conflict of interest.

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
