# Peer review of "Geomorphology, Geoarchaeology, and Geochronology of the Upper Pleistocene Archaeological Site of El Olivo Cave (Llanera, Asturias, Northern Spain)"

_geosciences, doi:10.3390/geosciences13100301_

Round 1

Reviewer 1 Report

This is a very interesting paper which show the interest of the geoarchaeological approach for understanding caves deposits and human settlements. The methods are useful and described in detail. They provide lot of results which support the interpretations. This is an example of what could be done in numerous pleistocene and holocene sites, highlighting the interest of interdisciplinary approaches.

Event if I am not a native english speaker, it appeas to me that the paper is clear and well written. Only a few remarks are reported in the attached PDF. The caption of one figure is still in spanish. Please translate it too.

Author Response

All changes suggested by the reviewer have been made.

Reviewer 2 Report

Dear editor,

This work concerns the geomorphological and geoarchaeological reconstruction of the upper Pleistocene history of a cave in Northern Spain. The methods used include standard sedimentological, soil and palaeopedological analysis as well as geomorphological survey. Dating is carried out through several different techniques which relate to different materials found inside the cave environment. This work is well carried out and collects a substantial amount of data. The information is explained and interpreted in a non-banal and effective way. All considered, the manuscript is well written and deserves publication. However, there are several concerns that lead me to suggest some revisions to the text.

Section 3 “Geomorphology” is almost non-existent and mostly deals with the stratigraphic setting of the cave and possible provenance of the sediments. On the other hand, the extensive model of landscape evolution given in section 6.1 gives a detailed historical reconstruction of erosion and of the general geomorphological processes in time. I believe that section 3 should be expanded with a detailed description of the current landforms and processes acting on the landscape in order to give the reader a full picture of the evolution of the area.

Section 4 “Geoarchaeology” reports all results without ever commenting on them. I believe that section 4.2 should contain a short summary for each subsection (especially 4.2.1, but also 4.2.2, 4.2.3, 4.2.4) exposing which data are relevant for the reconstruction in 6.2 and why. Please also include a table with all samples and analytical results, the use of graphs is not sufficient to share data.

The relationship with archaeological data is also not clear at all. The only piece of information comes from the presence of scarce lithics in level OL.7 attributed to the Middle Palaeolithic and of Magdalenian lithics and faunal remains from level OL.4 upwards. I believe that a wider description of the archaeological assemblage should be provided for each level, or at least as a scheme of the archaeological findings within the stratigraphy with relative dating.

I found the naming of levels and samples very confusing.

-          The dating samples share essentially the same name with levels they do not belong to. I would suggest either to rename them or to put some suffixes that clearly refer them as dating samples. I would also add a column containing the provenance of each sample in table 1

-          The external levels are labelled in Roman numerals in the text and Arab numbers in the figures and graphs, please use consistent naming

-          What is sample OL.7 Ox? Is it the same as OL.7b Mn? The same is valid for OL.4 gravas/OL.4 gravels and OL.6. Clays/OL.6 Arcillas, please be consistent throughout the text.

-          I believe the authors should include OL.4 gravas/OL.4 gravels and OL.6. Clays/OL.6 Arcillas in figure 7, in order for the reader to correlate the stratigraphic section with the results from the graphs (figure 10 to 22)

Section 6 gives the main interpretations and formulates a general model for the evolution of the landscape and the formation of the cave infilling.

Firstly, I believe that sections 6.1 and 6.2 should be inverted. As it is, the manuscript misses a level of explanation and jumps from the exposition of analytical data straight to discuss the landscape evolution model, completely missing the contribution of the sedimentological, mineralogical and edaphological analyses which appears only afterwards. Also, the paleogeographic reconstruction is informed by the geoarchaeological interpretation, while the opposite is much less so: for example, without the interpretation of fluvial dynamics in lines 644 onwards, phases 2 and 3 of section 6.1 are completely unjustified. It makes sense that the deposition of the cave infilling would be the proxy for the evolution of the landscape, and not vice versa.

From a more “narrative” point of view, I feel that this section is very limited to the explanation of each single piece of information and less about the general picture. In the end, the authors draw all the dots but do not connect them to show how the history of the cave informs about the relationship between landscape change, climate variations, and human agency (for example, it is not clear if there is a link between climate stages and fluvial dynamics). I think that the geoarchaeological interpretation should focus on the reconstruction of the events recorded in the cave and relate them to climate stages and human presence. The palaeogeographic model should instead provide a more general overview of the interaction between the various agents and show in which way climate and humans acted in the landscape (some of these observations are currently at the end of section 6.2 and in section 7).

Section 7 “Conclusions” provides a very useful summary of the history of the cave (much clearer than the chronology explained in section 6) and formulates hypotheses on the archaeological implications of the findings. I believe that this should belong to the main discussion instead, and that the conclusions should draw on the observations already made instead of providing new info (such as line 736). In particular, the archaeological remarks seem to appear out of nowhere, since the rest of the manuscript never touches the subjects of settlements and social space, nor makes comparisons with other relevant chronologies for the same period. I would suggest discussing these points in the previous section first, and only use the Conclusions to highlight them.

Some other remarks:

Line 55 – ancient times is not proper, please be more specific

Lines 170-171 – Please explain in detail the location of these strata and the relation with the rest of the cave in section 4.1. This is very important as it is the main age constraint for the whole sequence, so it should be related strongly to the rest of the cave environment, and not just a small note inside the Methods.

Lines 388-393 – this paragraph is repeated below

Lines 485-492 - F families are not explained or justified: why do the G-A, G-B1 and G-B2 families are similar in the fine fraction? Why is G-B3 different? What does this mean from a sedimentological point of view, and what does it imply about provenance and the nature of the sediment?

Lines 584-625 – Figure 23 is referred in the text as figure 22, please change

Lines 626-632 – This paragraph is unclear, and it feels like it should belong in the Introduction, please revise

Lines 648-651 – This is debatable, since the analytical results point to the opposite: the two deposits apparently belong to the same process and similar environmental conditions. The only difference in the two might be in stratigraphic position, but the two deposits are not located on the stratigraphy and in the figures. Please provide more evidence or avoid drawing conclusions not clearly supported by data.

Line 655 – the term “near-complete” implies that some remaining levels from this period have been found. Where are they located in the stratigraphy?

Lines 660-661 – The text implies that the formation of this small terrace (and consequently the exterior sequence) is earlier than the deposition of the excavated interior sequence. If so, what is its topographic relation with the rest of the deposits? Level OL.7 is below the sediments of the small terrace (as per figure 4) so it should be the oldest. Please explain

Line 715 – this section should be 7, please correct

Figure comments:

Figure 5 – Please use English captions

Figure 10, 11, 19, 20 – colours for the graphs are often very similar and difficult to visually compare, please improve their readability

Figure 10 and 11 – These graphs could be merged together in a single figure

Figure 12B – not all samples are named, please add information for each dot. The graph shows a distribution of values between lower and higher sand content. Is that a trend or not? Impossible to say without sample names.

Figures 13-18 – curve names are too small to read, could the authors provide a legend instead?

Figure 14 is missing, please add it

Figure 19 - I believe that for the main stratigraphies (interior and exterior) you could show data as line depth profiles, much like figure 10. Horizontal cumulative bars are very difficult to read.

Figure 20 and 22 – These graphs could be merged together in a single figure. I also believe that for the main stratigraphies (interior and exterior) you could show data as different line depth profiles, much like figure 10. Horizontal aligned bars are very difficult to read.

Figure 23 – in the caption is present a 5th stage named E which is not visible in the figure, please correct

Author Response

The suggestions and comments of reviewer 2 have been addressed as much as possible.

Round 2

Reviewer 2 Report

I have read the revised manuscript, and I believe that the text has improved by the changes made by the authors. Most of the critical issues I found are solved, albeit in some cases in a very laconic fashion.

I maintain my doubts on the changes to graphs I suggested that were not addressed by the authors, but they mostly involve readability and are not formally wrong.

Therefore, I do not find concerning obstacles to the publication of the manuscript as it is.

Author Response

Some of the figures indicated by reviewer 2 have been improved, reducing their number.

Thank you very much for not putting obstacles to the of publication of the manuscript after its review.
